# Shared neutrophil and T cell dysfunction is accompanied by a distinct interferon signature during severe febrile illnesses in children

Severe febrile illnesses in children encompass life-threatening organ dysfunction caused by diverse pathogens and other severe inflammatory syndromes. A comparative approach to these illnesses may identify shared and distinct features of host immune dysfunction amenable to immunomodulation. Here, using immunophenotyping with mass cytometry and cell stimulation experiments, we illustrate trajectories of immune dysfunction in 74 children with multi-system inflammatory syndrome in children (MIS-C) associated with SARS-CoV-2, 30 with bacterial infection, 16 with viral infection, 8 with Kawasaki disease, and 42 controls. We explore these findings in a secondary cohort of 500 children with these illnesses and 134 controls. We show that neutrophil activation and apoptosis are prominent in multi-system inflammatory syndrome, and that this is partially shared with bacterial infection. We show that memory T cells from patients with multi-system inflammatory syndrome and bacterial infection are exhausted. In contrast, we show viral infection to be characterized by a distinct signature of decreased interferon signaling and lower interferon receptor gene expression. Improved understanding of immune dysfunction may improve approaches to immunomodulator therapy in severe febrile illnesses in children.

Severe febrile illnesses in children requiring hospitalization and organ support, arise from diverse infections or inflammatory triggers, and overlap with the dysregulated host response that characterizes sepsis[1–3]. Diagnosing the etiology of febrile illness is difficult due to shared clinical features and time-consuming microbiological tests, hindering prompt initiation of pathogen-specific treatment. In contrast, comparative immunology of these illnesses may describe shared and distinct host immune features. Beneficial use of immunomodulators for COVID-19[4,5] and multi-system inflammatory syndrome in children (MIS-C)[6–10], highlight the potential for treating immune dysregulation. Although the incidence of MIS-C has now declined, likely because of natural and vaccine-induced population immunity against SARS-CoV-2, a comparison of MIS-C with other severe febrile

illnesses provides a remarkable opportunity to understand mechanisms of disease and move towards host-directed immunomodulation.

MIS-C emerged in the early months of the COVID-19 pandemic as a life-threatening childhood illness following SARS-CoV-2 exposure and is characterized by fever, rash, conjunctival injection, severe cardiac dysfunction, multi-organ involvement, and intense inflammation[11,12]. Clinically, MIS-C shares similarities with severe bacterial infection (SBI) including toxic shock syndrome (TSS), and Kawasaki disease (KD)[13]. Both MIS-C and KD can also cause coronary artery aneurysms (CAA). MIS-C may share immunological features with SBI and KD including a skew towards activated immature neutrophil populations, markers of neutrophil extracellular trap (NET) production, and reduced markers for antigen presentation[14–19]. MIS-C and TSS are associated with

✉e-mail: manu.shankar-hari@ed.ac.uk; m.levin@imperial.ac.uk

preferential usage of disease-specific TCR-Vβ variables genes in activated CD4[+] and CD8[+] T cells, suggesting a super-antigenic phenomenon[17,20,21] that may contribute to T cell apoptosis-driven lymphopenia in MIS-C[14]. We therefore hypothesized that MIS-C would be associated with more profound T-cell exhaustion and apoptosis than SBI. Finally, severe viral infections (SVI), including adult COVID-19 pneumonitis, are associated with altered type I interferons responses[22,23], while MIS-C, SBI, and KD are associated with altered type II IFN signaling[14,24,25]. We therefore compared IFN and other cytokine production, and phosphorylation of IFN signal transduction and activator of transcription (STAT) and other transcription factors, between these illnesses.

To investigate these hypotheses, we used extensive immunophenotyping by mass cytometry[26], and cell stimulation with supernatant cytokine assays. We used principal component analysis and hierarchical clustering to identify clusters of patients with similar immune features, regardless of etiology of severe febrile illness[27]. We used unsupervised multi-omic factor analysis (MOFA)[28] to probe latent factors within the mass cytometry data between illnesses, and across the trajectory of illness from acute presentation to convalescence. We subsequently used gene expression data from whole blood RNA sequencing (RNA-Seq) in a separate Transcriptomic Cohort of severe febrile illness in children[29] to explore our findings (Fig. 1a).

## Results
### Characteristics of the study cohorts
We recruited 137 children (Derivation Cohort) admitted to hospital with the following severe febrile illnesses: MIS-C ($n = 74$), SBI ($n = 30$), SVI ($n = 16$), KD ($n = 8$) and other severe inflammatory illnesses ($n = 9$). Each child was sampled at up to three time points across their disease trajectory: acute presentation (T1), defervescence (T2) and convalescence (T3), as reported previously[13] (Fig. 1b, and Supplementary Fig. 1). We also recruited 15 approximately age- and sex-matched healthy pediatric controls (HPC; 15/15 unvaccinated for SARS-CoV-2, 1/15 SARS-CoV-2 seropositive; Table 1) and 27 healthy COVID-19 vaccinated adult controls (HAC; Supplementary Table 1). All participants in this Derivation Cohort were recruited between July 2020 and May 2022. We also analyzed RNA-seq data from a previously reported cohort of 372 children (Transcriptomic Cohort) with severe febrile illness (MIS-C, $n = 38$; bacterial, $n = 188$; viral, $n = 138$; KD, $n = 136$). The patients in the Transcriptomic Cohort were recruited between August 2003 to April 2021 (full details are provided in Jackson et al.[29]). The Derivation and Transcriptomic Cohorts had no overlap.

The characteristics of the Derivation Cohort are shown in Table 1. The children were diagnosed using standard clinical criteria[29,30] and represented severe febrile illness in children during the COVID-19 era. As such, 55/74 (74%) children with MIS-C, 25/30 (86%) children with SBI, 14/16 (88%) children with SVI, 0/8 children with KD and 3/9 (33%) children with other inflammatory disease (two children with autoimmune encephalitis, two juvenile idiopathic arthritis, two myocarditis, one macrophage activation syndrome, one reactive arthritis, one asthma) required admission to intensive care units. Eight (11%) children with MIS-C, and two (25%) children with KD developed CAA. Sixty-seven (91%) children with MIS-C had positive SARS-CoV-2 spike IgG and 10 children were SARS-CoV-2 polymerase chain reaction (PCR) or antigen positive (MIS-C 3/74, 4%; SBI 1/30, 3.3%; SVI 6/16, 38%). Children in the Transcriptomic Cohort were diagnosed using the same standard clinical criteria[31]. Characteristics of HAC and the Transcriptomic Cohort are shown in Supplementary Tables 1 and 2, respectively. The number of samples at different timepoints analyzed with mass cytometry and cell stimulation experiments are depicted in Fig. 1b, while the overlap of samples across experiments is shown in Supplementary Fig. 2.

### Immunophenotyping of peripheral blood
We developed a 43-marker mass cytometry panel staining for CD45 "barcoding"[25], cell surface proteins, intracellular cytokines, and phosphorylation of nuclear transcription factors and measured the proportions of immune cell subsets and marker expression on 139 whole blood samples (Supplementary Fig. 3a–g). These included 77 samples (T1:35, T2:16, T3:26) from 38 children with MIS-C, 31 samples (T1:17, T2:8, T3:6) from 17 children with SBI, 12 samples (T1:10, T2:2) from 10 children with SVI 8 samples (T1:4, T2:1, T3:3) from 4 children with KD, and 10 samples from 10 HPC (Supplementary Table 3). A proportion of T1 samples were post-treatment: MIS-C 17 (49%) post corticosteroids and 5 (17%) post immunoglobulins; SBI, 2 (12%) post immunoglobulins; KD 2 (50%) post corticosteroids and 3 (75%) post immunoglobulins; SVI 0 post corticosteroids.

We first examined the proportions of all immune cell subsets and marker expression (collectively, "immune features") at T1 using principal component analysis (PCA; Fig. 2a and Supplementary Fig. 4a–c). Febrile illness groups were separate from a distinct group of HPCs in dimension 3. There was considerable overlap in febrile illness groups on PCA suggesting shared immune features. The centroid of the MIS-C cluster extended along dimension 1, where the dominant eigenvectors were increased intracellular IL6 expression in memory CD4[+] T cells, and phosphorylated nuclear factor kappa B p65 (NFκB) expression in mature neutrophils and CD95/Fas expression in immature neutrophils in dimension 3. The centroid of the SBI cluster was closely aligned with that of the MIS-C cluster, but typically closer to the PCA plot origin, suggesting less extreme immune features. The centroid for SVI cluster was most distinct from other groups in dimension 2 and dimension 3. Intriguingly, staining for γδTCR was identified on mature neutrophils, possibly indicating an interaction between γδT cells and neutrophils.

To identify distinct immune features of each disease group we used general linear models (GLM) in T1 samples from MIS-C, SBI and SVI and SBI/SVI combined (Fig. 2b). Immune feature values were scaled to the median values of HPC for each feature, with subsequent Benjamini-Hochberg (BH) adjustment for multiple comparisons (Fig. 2b). KD was not included in this analysis due to low sample size ($n = 4$). Of the 37 immune features that were differentially expressed between groups, 36 were between MIS-C and SBI/SVI combined. The GLM identified three themes distinct to MIS-C: 1) activated/apoptotic innate immune cells, 2) dysregulated T cell function, and 3) mature and activated myeloid dendritic cells (mDCs) and plasmacytoid (pDCs). This analysis was supported by hierarchical clustering that identified two clusters of patients in the Derivation Cohort (Fig. 2c): Cluster 1 contained 29 children at T1 (22, 76%, MIS-C; 4, 14%, SBI; 3, 10%, SVI) and Cluster 2 contained 37 children at T1 (13, 35%, MIS-C; 13, 35%, SBI; 7, 19%, SVI; 4, 11%, KD). Cluster 1 contained the majority of MIS-C cases and children in Cluster 1 were older (median 9.3, IQR 4.9–14, years versus 5.4, IQR 1.6–8.9, years, $p < 0.001$), and more likely to receive vasoactive infusions (19, 66%, children versus 15, 41%, children, $p = 0.051$) in comparison with Cluster 2 (Supplementary Table 4). Cluster 1 was characterized by higher CD64 (FCγRI) expression on monocytes and mature neutrophils, and expression of exhaustion (CD279/PD1) and elevated apoptotic (CD95/Fas) markers on memory CD4[+] and CD8[+] T cells and innate immune cells (neutrophils, natural killer (NK) cells, and monocytes) in comparison with Cluster 2. Cluster 1 also displayed lower HLA-DR and CD11c on mDCs, and lower expression of CD278, an inducible T-cell co-stimulator, on monocytes, mDCs, pDCs and mature neutrophils and lower IL17Ra on memory T$_{reg}$ cells in comparison with Cluster 2. Children in Cluster 1 therefore show a more extreme immunophenotype than children in Cluster 2. Cluster membership was not associated with immunomodulator treatment prior to acute sampling (Supplementary Fig. 5).

To extend our analysis of the immune cell changes over the disease trajectory in an unsupervised manner, we used multi-omic factor

analysis (MOFA)[28]. This is a computational method used to discover the principal axes of variability within high-parameter datasets, and the resulting factors can be interpreted like a principal component analysis. We also explored membership of the previously identified Cluster 1 and Cluster 2 in the context of this analysis. MOFA revealed four Factors which collectively accounted for 81.5% of the variability in the dataset (Supplementary Fig. 6). Factor 1 accounted for 36% of the variability within the data and represented myeloid cell activation and T cell activation. Factor 1 was associated with CD64 and CD95/Fas upregulation in neutrophils, NK cells, monocytes, and CD38 upregulation on memory CD4$^+$ and CD8$^+$ and γδT cells. Factor 1 scores were positively associated with MIS-C at T1 and with Cluster 1 membership;

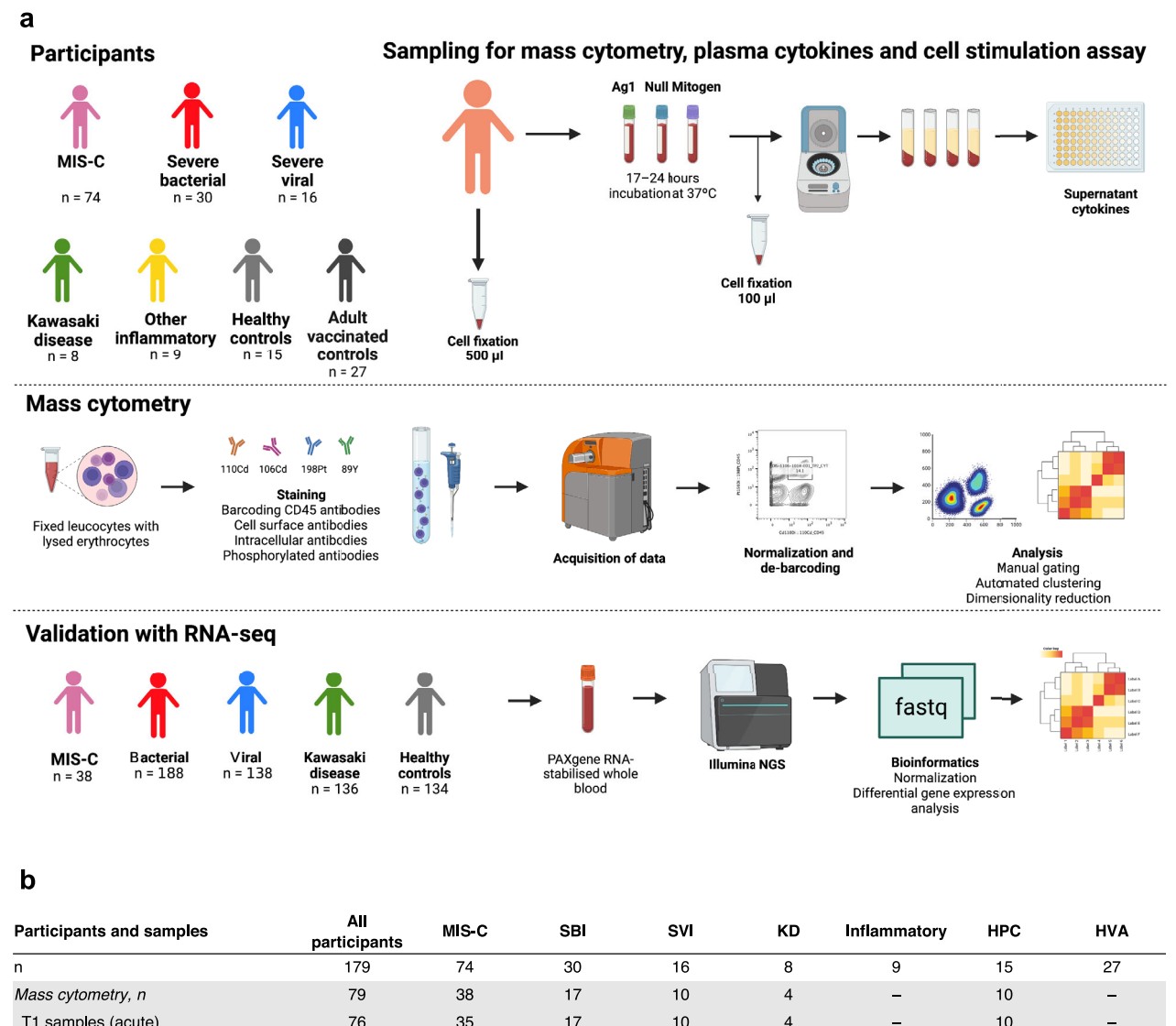

**b**

| Participants and samples | All participants | MIS-C | SBI | SVI | KD | Inflammatory | HPC | HVA |
|---|---|---|---|---|---|---|---|---|
| n | 179 | 74 | 30 | 16 | 8 | 9 | 15 | 27 |
| *Mass cytometry, n* | 79 | 38 | 17 | 10 | 4 | – | 10 | – |
| T1 samples (acute) | 76 | 35 | 17 | 10 | 4 | – | 10 | – |
| T2 samples (defervescence) | 27 | 16 | 8 | 2 | 1 | – | – | – |
| T3 samples (convalescence) | 35 | 26 | 6 | 0 | 3 | – | – | – |
| *Cell stimulation: supernatant cytokines, n* | 131 | 52 | 16 | 7 | 4 | 9 | 15 | 27 |
| T1 samples (acute) | 74 | 41 | 14 | 7 | 1 | 9 | – | – |
| T3 samples (convalescence / controls) | 66 | 17 | 3 | 1 | 3 | 0 | 15 | 27 |
| *Cell stimulation: mass cytometry, n* | 15 | 12 | – | – | – | – | 3 | – |
| T1 samples (acute) | 12 | 12 | – | – | – | – | – | – |
| T3 samples (convalescence / controls) | 9 | 6 | – | – | – | – | 3 | – |

Abbreviations: MIS-C, multi-system inflammatory syndrome in children; SBI, severe bacterial infection; SVI, severe viral infection; KD Kawasaki disease; HPC, healthy pediatric control; HVA, healthy vaccinated adult. n refers to number of participants (some of whom had multiple sample-time points.

**Fig. 1 | Schematic representation of the study cohort, experimental flow and samples analyzed. a** Summary of the number of samples used for mass cytometry and cell stimulation assay from each disease groups at timepoints T1 (acute), T2 (defervescence), T3 (convalescence). **b** Sample numbers available and analyzed. For a full overview of the sample types and at an individual level see the Supplementary

Fig. 1. Abbreviations: MIS-C, multi-system inflammatory syndrome in children; SBI, severe bacterial infection; SVI, severe viral infection; KD, Kawasaki disease; HPC, healthy pediatric control; HAC, healthy adult control. Figure 1a created with BioRender.com released under a Creative Commons Attribution-NonCommercial-NoDerivs 4.0 International license. Source data are provided as a Source Data file.

**Table 1 | Demographic, clinical and laboratory features of patients recruited to the Derivation cohort in the study**

| Characteristics | MIS-C | SBI[a] | SVI[b] | KD | Other Inflammatory | HPC |
|---|---|---|---|---|---|---|
| *n* | 74 | 30 | 16 | 8 | 9 | 15 |
| Age (years) (median, IQR) | 9.6 (5.4–13.2) | 9.6 (4.4–14) | 2.9 (1.7–13) | 1.4 (0.8–4.6) | 10.7 (5.5–13) | 6.6 (5.1–10) |
| Female (*n*, %) | 27 (36%) | 12 (41%) | 9 (56%) | 2 (25%) | 3 (33%) | 6 (40%) |
| *Self-reported ethnicity* | | | | | | |
| Asian (*n*, %) | 10 (14%) | 6 (20%) | 4 (25%) | 4 (50%) | 3 (33%) | 0 |
| Black (*n*, %) | 15 (21%) | 1 (3.3%) | 3 (19%) | 0 | 1 (7.7%) | 3 (20%) |
| Other (*n*, %) | 19 (26%) | 3 (10%) | 0 | 1 (13%) | 1 (11%) | 2 (13%) |
| White (*n*, %) | 28 (39%) | 20 (67%) | 9 (56%) | 3 (38%) | 5 (56%) | 10 (67%) |
| Comorbidity[c] | 8 (11%) | 7 (23%) | 2 (13%) | 2 (25%) | 1 (11%) | 0 |
| *Presenting clinical features* | | | | | | |
| Gastrointestinal symptoms | 60 (81%) | 20 (67%) | 6 (38%%) | 6 (75%) | 3 (33%) | – |
| Oral mucositis | 7 (9.4%) | 1 (3.3%) | 0 | 3 (38%) | 0 | – |
| Conjunctivitis | 40 (54%) | 2 (6.7%) | 0 | 6 (75%) | 1 (11%) | – |
| Rash | 43 (58%) | 10 (33%) | 2 (13%) | 7 (88%) | 4 (44%) | – |
| Respiratory symptoms | 11 (15%) | 17 (57%) | 12 (75%) | 0 | 4 (44%) | – |
| *Additional findings during admission* | | | | | | |
| Max CRP concentration (mg/l) (median, IQR) | 218 (151–270) | 190 (146–293) | 72 (17–149) | 140 (90–166) | 66 (31–149) | – |
| Max neutrophil count (×10⁹/l) (median, IQR) | 12 (8.0–16) | 15 (9.0–24) | 8.6 (6.9–12) | 12 (7.4–16) | 15 (9.2–20) | – |
| Min lymphocyte count (×10⁹/l) (median, IQR) | 0.6 (0.4–1.0) | 1.3 (0.6–2.4) | 1.1 (0.7–2.0) | 2.5 (2.0–3.2) | 1.1 (1.0–1.3) | – |
| Coronary artery aneurysm (*n*, %) | 8 (11%) | 0 | 0 | 2 (25%) | 0 | – |
| *Treatments during admission* | | | | | | |
| Mechanical ventilation (*n*, %) | 4 (5.4% | 14 (50%) | 13 (81%) | 0 | 2 (25%) | – |
| Vasoactive infusion (*n*, %) | 49 (67%) | 14 (50%) | 5 (33%) | 0 | 1 (13%) | – |
| Intravenous immunoglobulin | 20 (27%) | 3 (10%) | 0 | 5 (63%) | 1 (11%) | – |
| Before T1 sampling | 12 (16%) | 2 (6.7%) | 0 | 0 | 0 | |
| High dose corticosteroids | 52 (70%) | 1 (3.3%) | 1 (6.3%) | 4 (50%) | 3 (33%) | – |
| Before T1 sampling | 33 (34%) | 0 | 1 (6%) | 3 (60%) | 3 (33%) | |
| Monoclonal antibody therapy[d] | 17 (24%) | 0 | 0 | 1 (13%) | 0 | – |
| *Outcome* | | | | | | |
| PICU admission (*n*, %) | 55 (74%) | 25 (86%) | 14 (88%) | 0 | 3 (33%) | – |
| Hospital length of stay (days) (median, IQR) | 6.5 (5.2–8.6) | 8.6 (6.7–23.2) | 14 (11–26) | 5.4 (4.1–8.6) | 8.6 (8.3–9.2) | – |
| In-hospital mortality (*n*, %) | 1 (1.4%) | 2 (6.7%) | 1 (6.3%) | 0 | 0 | – |
| *Infections identified* | | | | | | |
| SARS-CoV-2 PCR/antigen positive (*n*, %) | 3 (4.0%) | 1 (3.3%) | 6 (38%) | 0 | 0 | – |
| SARS-CoV-2 IgG antibody positive (*n*, %) | 67 (90.5%) | 5 (17%) | 6 (38%) | 1 (13%) | 0 | 1 (6.7%) |

*SBI* severe bacterial infection, *SVI* severe viral infection, *KD* Kawasaki disease, *HPC* healthy pediatric control.

[a]Children had the following identified by culture of normally sterile site: *Staphylococcus aureus* (5), Group A streptococci (3), non-typhoidal *Salmonella* spp. (3), *Escherichia coli* (2), *Pseudomonas aeruginosa* (2), *Streptococcus pneumoniae* (2), other streptococci (2), *Abiotrophica defectiva* (1), *Enterococcus faecalis* (1), *Haemophilus parainfluenzae* (1); culture negative: toxic shock (4), septic shock with turbid pericardial effusion (1), meningitis with CSF neutrophilia (1), fecal peritonitis and shock (1).

[b]Children had the following detected by PCR and attributed to etiology: SARS-CoV-2 (7), rhinovirus (3, 2 as co-infections with other viruses), enterovirus (2), parainfluenzae viruses (2), respiratory syncytial virus (2), Epstein-Barr virus (1), parechovirus (1).

[c]MIS-C: Chronic respiratory disease (3), neurological/psychological disorder (3), previous severe inflammatory disorder (2), immunosuppression with anti-inflammatory medication (1), endocrinological disorder (1 child). Severe bacterial infection (SBI): Neurological/psychological disorder (3), chronic respiratory disease (3), cardiac (1), endocrinological (1), other genetic syndrome (1 child); Severe viral infection (SVI): Neurological (2), other genetic syndrome (1 child); Kawasaki disease (KD): Cardiac (1), renal (1); Other inflammatory: Previous severe inflammation (1 child).

[d]MIS-C: tocilizumab (10), anakinra (3) infliximab (2), anakinra and tocilizumab (1 child); other inflammatory: anakinra (1), infliximab (1 child).

and negative associated with MIS-C at T3, KD at T3 and Cluster 2 membership (Supplementary Fig. 6). For MIS-C, there was a significant trend of decreasing Factor 1 scores between T1 and T3. For SBI, there was a non-significant trend of decreasing Factor 1 scores between T1 and T3 (Supplementary Fig. 7a–i). Factor 1 scores were positively associated with severity of inflammation (as determined by C-reactive protein concentration[32], adjusted $R^2 = 0.636$, $p = 0.024$), and children with MIS-C had higher Factor 1 scores relative to children SBI and KD independent of CRP concentration ($p = 0.004$ and $p = 0.020$ respectively; Supplementary Fig. 7j).

Factor 2 represented restoration of innate and adaptive immune cells to baseline, with a decrease in the proportion of immature neutrophils, increased CD11b and CD16 expression and decreased phosphorylation of NFκB in myeloid cells, and increased CD28 changes in T cells, among other changes (Supplementary Fig. 8). Factor 2 scores were significantly higher in SVI T1 compared with MIS-C T1 and SBI T1 and increased significantly between MIS-C T1 to MIS-C T3 (Supplementary Figs. 6, 7). Factor 3 represented decreased phosphorylation of the cell signaling molecules STAT1 and STAT5 and NFκB. Factor 3 scores were not significantly different at T1 between any illness

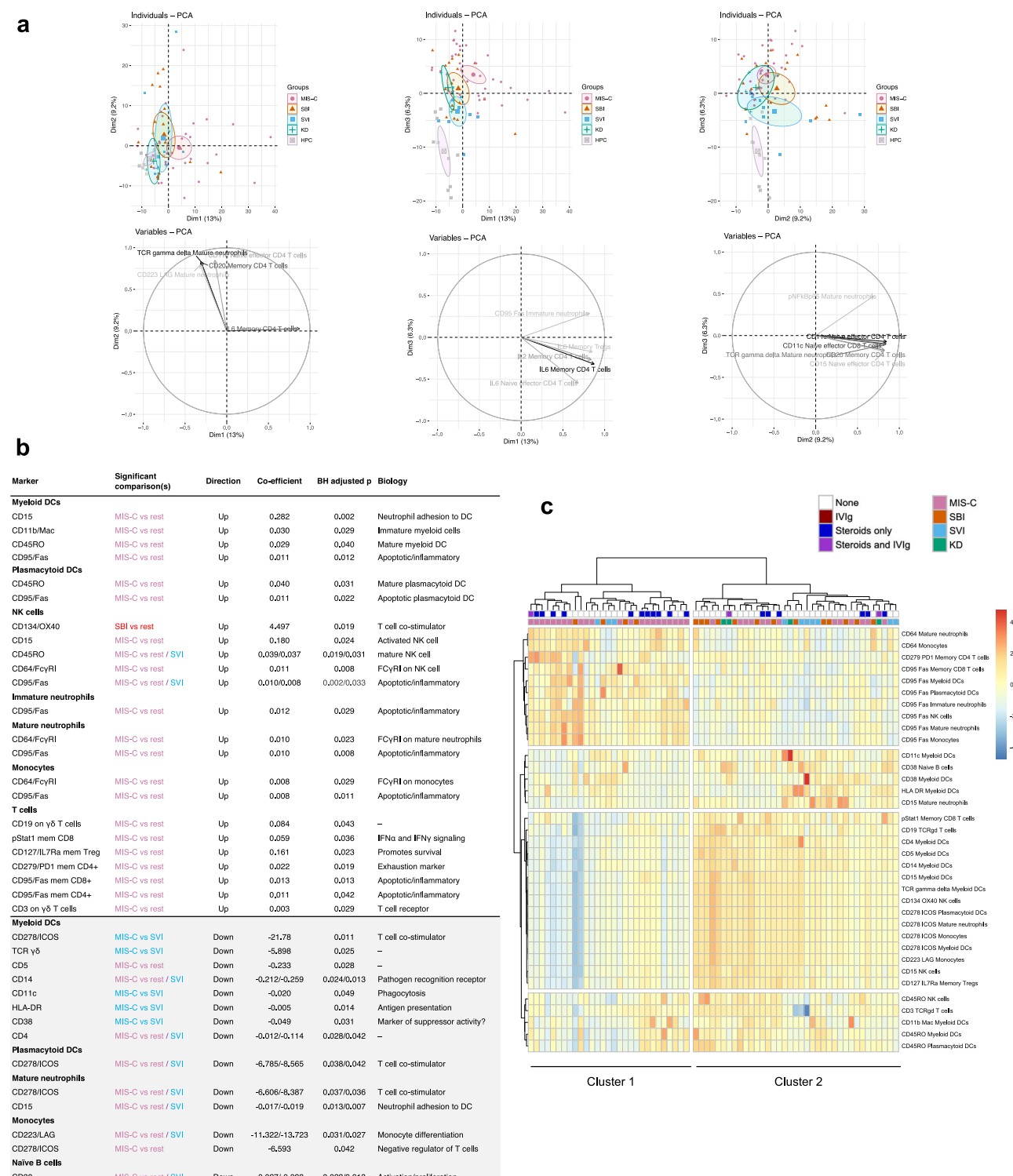

**Fig. 2 | Supervised and unsupervised approaches to immune cell data show differences between disease groups and key immune cell features in acute (T1) disease. a** Principal component analyses of immune cell features measured by patient and disease group (small symbol; upper) and eigenvectors (lower; top 5 eigenvectors shown). Ellipses indicate 95% confidence intervals around the group center (large symbol). **b** Results of a generalized linear model regressing immune cell features with each disease group. Comparisons included MIS-C vs SBI and SVI combined ("rest") as well as at an individual group level. The Benjamini-Hochberg method was used to adjust the two-sided *p* value for multiple testing; only comparisons with adjusted *p* < 0.05 have been shown. **c** Heatmap displaying immune cell features associated with disease group (**b**) with disease groups clustered as columns (Ward's hierarchical clustering), and data scaled by row. Source data are provided as a Source Data file.

groups, but decreased between MIS-C T1, and T3 (Supplementary Figs. 6 and 7). Factor 4 represented decreased myeloid cell production of IL2 and was negatively associated with SVI T1, indicating increased myeloid cell production in SVI T1 relative to other illness groups (Supplementary Fig. 6). Taken together, MOFA showed myeloid cell and T cell activation markers dominate the differences between groups of patients with severe febrile illnesses, and innate and adaptive immune cell function is restored by convalescence in MIS-C and SBI. Markers of T cell activation, exhaustion and apoptosis were significantly correlated in CD4 memory T cells in particular (Supplementary Figs. 9–10). We therefore used cell stimulation experiments to investigate whether T cell function is a key feature of MIS-C immunopathology.

### Immune cell function: cell stimulation experiments

We compared the functional cellular immune responses between illness groups and across timepoints. Fresh whole blood was incubated with either SARS-CoV-2 receptor binding domain (RBD) spike antigen peptides (Ag1), phytohemagglutinin (a non-specific T cell stimulant, mitogen), or unstimulated negative control (nil) for 17–24 h (Quanti-FERON SARS-CoV-2 RUO assays, Qiagen) prior to harvesting supernatant for multiplexed cytokine quantification (13-plex cytokine panel, LegendPLEX, Biolegend).

These assays were performed on a total of 131 acute (T1) and convalescent (T3) samples: 58 samples (T1:41, T3:17) from children with MIS-C, 16 samples (T1:13, T3:3) from children with SBI, 8 samples (T1:7, T3:1) from children with SVI, 4 samples (T1:1, T3:3) from children with KD, 9 samples (T1:9) from children with other inflammatory syndromes, 15 HPC samples, and 27 HAC samples. The summary cytokine results for each illness group and timepoints are depicted in Supplementary Table 5. A proportion of T1 samples were post-treatment: MIS-C, 20 (49%) post corticosteroids, 1 (2.4%) post IVIG, 7 (17%) post steroids and IVIG; SBI, 3 (23%) post steroids and IVIG; SVI 5 (71%) post steroids; KD 1 (100%) post steroids and IVIG and Inflammatory 2 (22%) post steroids, 1(11%) post IVIG and 1 (11%) post IVIG and steroids. There was no effect of treatment on supernatant cytokine concentration (Supplementary Fig. 11).

In supernatant from cells with no stimulation, IFNγ inducible protein 10 (IP10) was increased in MIS-C T1 and SVI T1 compared with the other inflammatory disease group T1, HPC and HAC (Fig. 3a). IL10 was raised in MIS-C T1 compared to HPC, and raised in MIS-C T1, SBI T1, SVI T1 and HPC compared to HAC. IL8 was reduced in MIS-C T1 compared with SBI T1, other inflammatory group T1 and HAC (Supplementary Fig. 13). These differences were not present in the convalescent samples and no differences were observed in all other cytokines tested (Fig. 3a, Supplementary Fig. 12a and 13a).

Stimulation with Ag1 and measurement of IFNγ showed the percentage increase in IFNγ concentration to be significantly lower in MIS-C T1, SBI T1 and HPC compared to HAC, with recovery in upregulation of IFNγ concentration seen in MIS-C T3 and SBI T3 (Fig. 3b). A similar trend was for IL6, IP10 and IL10 with incomplete recovery in convalescence (Fig. 3b and Supplementary Fig. 12b). The percentage increase in other cytokines following Ag1 stimulation remained similar across the illness groups and healthy control groups (Fig. 3b and Supplementary Figs. 12b and 13b).

Following stimulation with mitogen, there was a trend of lower upregulation of IFNγ and other proinflammatory cytokines (IL1β, IL6, IL10, IP10, TNFα, GM-CSF, IFNα2, IFNβ and IFNλ) in MIS-C T1, SBI T1 and SVI T1 compared to HPC with evidence of recovery at T3 (Fig. 3c, Supplementary Figs. 12c and 13c). These changes were not uniform across all groups, of note, concentrations of type I interferons (IFNα2 and IFNβ) were exclusively lower in SBI T1 compared with HPC (Supplementary Fig. 12c).

Taken together, the cell stimulation experiments demonstrate a diminished Ag1 response in acute MIS-C compared to vaccinated HAC

and a reduced response to mitogen across all acute febrile illness groups, and there is recovery in both responses in convalescence.

We sub-analyzed supernatant cytokines levels between Cluster 1 and Cluster 2 following cell stimulation (Supplementary Fig. 14). Cells in Cluster 1 produced higher IFNγ levels following mitogen stimulation ($p = 0.02$) in comparison with Cluster 2, suggesting that these Clusters may have functional importance.

### Immunophenotyping of cells from the stimulation assay.

We explored myeloid cell activation and impaired T cell function using samples following the cell stimulation assay from 12 children with MIS-C (T1:8, T3:6) and 3 HPC. Raw data were used for analysis of unstimulated samples, while foldchange from baseline was evaluated for stimulated samples.

Three distinct groups were observed on the innate immune response heatmap: (1) MIS-C T1, (2) MIS-C T3 and (3) HPC (Fig. 4a), while the following groups were observed on the T cell immune response heatmap: (1) MIS-C T1 (nil, Ag1, it), (2) MIS-C T3 (nil, Ag1) and HPC (nil, Ag1) and (3) MIS-C T3 (mit) and HPC (mit) (Fig. 4b).

In unstimulated samples, MIS-C T1 displayed innate immune cell inflammatory features: increased proportions of neutrophils, and non-classical monocytes compared to MIS-C T3 and HPC (Fig. 4a). Additionally, MIS-C T1 displayed reduced proportions of pDCs, mDCs, NK cells, and classical monocytes compared to MIS-C T3. Notably, CD95/Fas expression and intracellular IL17 and IL2 expression in innate immune cells was increased in MIS-C T1 compared to MIS-C T3 and HPC. Regarding the T cell immune response, MIS-C T1 exhibited lower proportions of naïve effector CD4+ T cells, memory CD4+ T cells, naïve effector CD8+ T cells, and γδT cells compared to MIS-C T3 and HPC (Fig. 4b). Of note, the proportion of all T cells expressing IL2 was highest in MIS-C T1, while the proportions of all T cells expressing IL6 was greatest in HPC. Most of the significant results from the pairwise comparisons related to differences observed between MIS-C T1, MIS-C T3 and HPC at baseline (Fig. 4c).

For samples stimulated with Ag1, the foldchange in proportions of activated innate cells was increased in MIS-C T1 compared with MIS-C T3 and HPC. Conversely, the foldchange in innate cells expressing intracellular cytokines was lower in MIS-C T1 compared to MIS-C T3 (Supplementary Fig. 15a). MIS-C T1 also demonstrated increased upregulation in proportions of all T cells, IL2 expressing T cells, as well as CD152 and CD95/Fas expressing naïve effector and memory CD4+ and CD8+ T cells.

Following mitogen stimulation, the overall response in HPC was highest, followed by MIS-C T3 and then MIS-C T1, particularly, in the proportions of activated innate immune cells and cells expressing IL17A and IL2 (Supplementary Fig. 15a). Of note, HPC had the smallest increase in proportion of T cells but exhibited the greatest increase in proportions of T cells expressing CD279, CD95/Fas, IL6 and TNFα (Supplementary Fig. 15b).

### Exploration of findings using whole blood gene expression analyses

We explored our hypotheses of myeloid cell activation, impaired T cell function and dysregulated interferon responses in the Transcriptomic Cohort. Gene set enrichment analysis[33] confirmed myeloid leukocyte activation, T cell differentiation and T cell activation and response to cytokines as key Gene Ontology pathways in MIS-C, bacterial infection and viral infection (Supplementary Fig. 16). We therefore performed targeted analyses of expression of genes associated with these pathways in whole blood using Wilcoxon testing with BH adjustment for multiple testing.

Myeloid cell activation is associated with FcγRI expression and NET production. We therefore explored genes associated with immunoglobulin receptor expression. The gene encoding FcγRI (*FCGR1A*) showed elevated expression in MIS-C, bacterial infection and KD, in

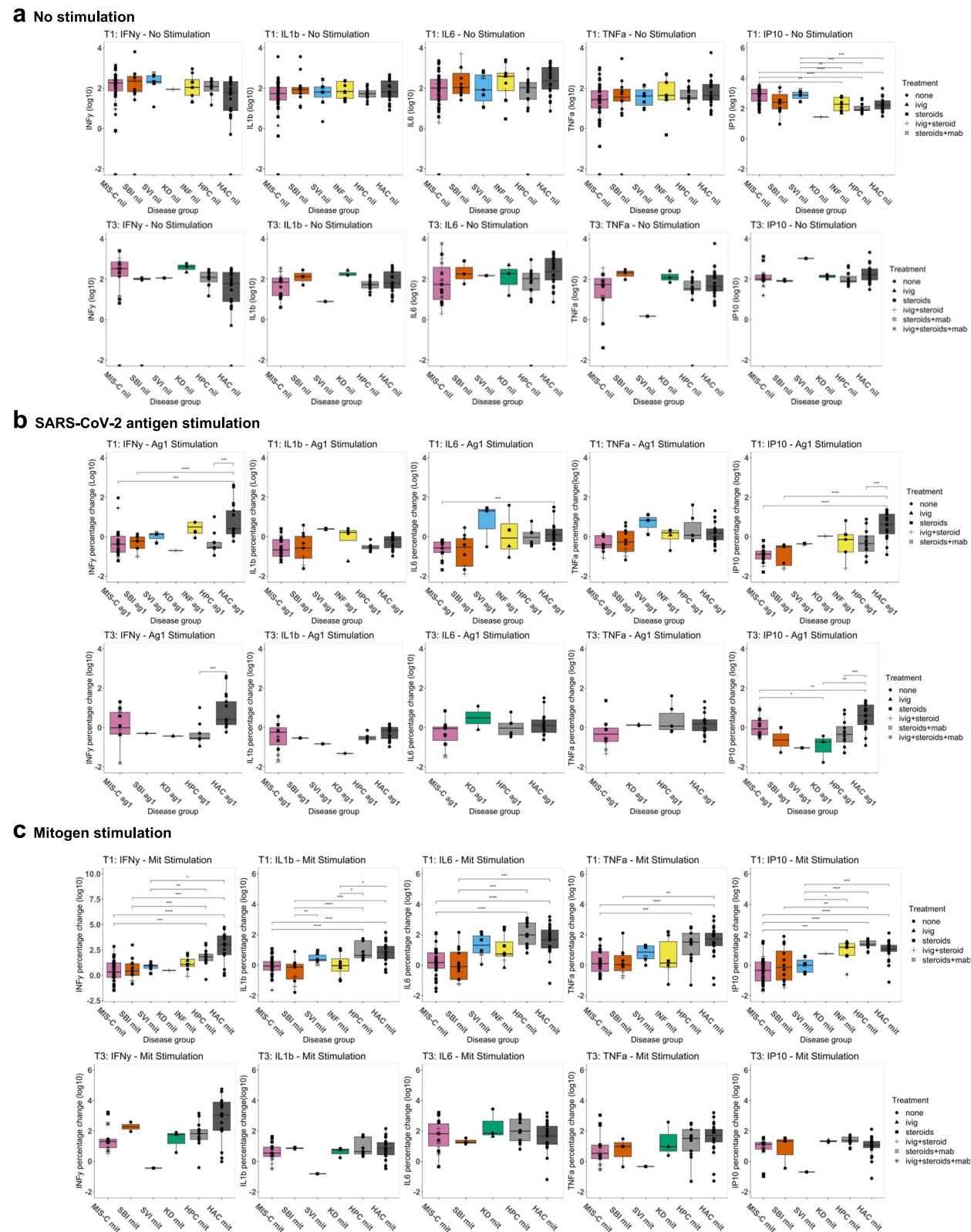

**Fig. 3 | Multiplex cytokine assay results of cell stimulation assay supernatant.** Concentration (picog/ml) or fold change of IFN, IFNβ, IL6, TNF, and IP10 in acute (T1) and convalescent (T3) samples with (**a**), No stimulation, (**b**), SARS-CoV-2 antigen stimulation and (**c**), mitogen stimulation. Log$_{10}$ transformed data are shown. Each point represents a participant for the following groups: multisystem inflammatory syndrome in children (MIS-C), severe bacterial illness (SBI), severe viral illness (SVI), Kawasaki disease (KD), other inflammatory disease (INF). Point shape corresponds to treatment received prior to blood sampling: ivig = intravenous immunoglobulin, steroid = corticosteroids, mab = monoclonal antibody. Two-sided Wilcoxon pairwise comparisons with Benjamini-Hochberg correction were performed; only significant *p* values are displayed. *=0.05, **=0.01, ***=0.001, ****=0.0001. Boxplots represent the median (horizontal line), first and third quartile boundary (box) and 1.5 times the inter-quartile range (whiskers). Source data are provided as a Source Data file.

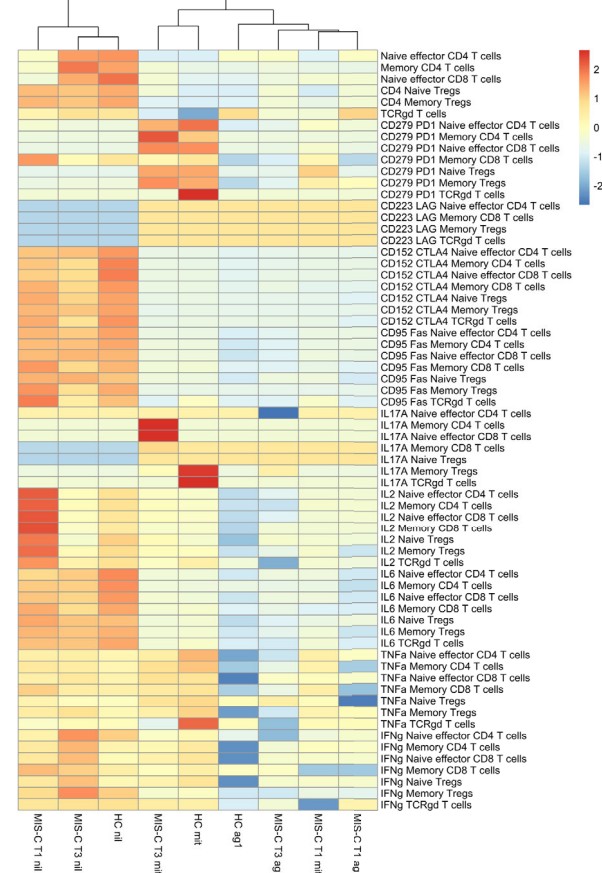

**a  innate immune response**

**b  T cell immune response**

**c**

| Stimulation status | Variable | Group 1 | Group 2 | p value | BH-adjusted p |
|---|---|---|---|---|---|
| | | **Innate immune response** | | | |
| No stimulation | CD95 Immature neutrophils | HPC | MIS-C T1 | 0.018 | 0.027 |
| No stimulation | CD95 Immature neutrophils | MIS-C T1 | MIS-C T3 | 0.008 | 0.023 |
| No stimulation | CD95 Mature neutrophils | MIS-C T1 | MIS-C T3 | 0.002 | 0.005 |
| No stimulation | CD95 Classical monocytes | HPC | MIS-C T1 | 0.018 | 0.027 |
| No stimulation | CD95 Classical monocytes | MIS-C T1 | MIS-C T3 | 0.003 | 0.008 |
| No stimulation | CD95 NK cells | HPC | MIS-C T1 | 0.009 | 0.014 |
| No stimulation | CD95 NK cells | MIS-C T1 | MIS-C T3 | 0.002 | 0.005 |
| No stimulation | CD45RO Myeloid DCs | HPC | MIS-C T1 | 0.018 | 0.027 |
| No stimulation | CD45RO Myeloid DCs | MIS-C T1 | MIS-C T3 | 0.001 | 0.002 |
| No stimulation | CD45RO NK cells | HPC | MIS-C T1 | 0.018 | 0.027 |
| No stimulation | CD45RO NK cells | MIS-C T1 | MIS-C T3 | 0.018 | 0.027 |
| | | **T cell immune response** | | | |
| No stimulation | Naïve effector CD4 T cells | HPC | MIS-C T1 | 0.016 | 0.024 |
| No stimulation | Naïve effector CD4 T cells | MIS-C T1 | MIS-C T3 | 0.006 | 0.017 |
| No stimulation | Memory CD4 T cells | HPC | MIS-C T1 | 0.009 | 0.014 |
| No stimulation | Memory CD4 T cells | MIS-C T1 | MIS-C T3 | 0.008 | 0.014 |
| No stimulation | Naïve effector CD8 T cells | HPC | MIS-C T1 | 0.009 | 0.014 |
| No stimulation | Naïve effector CD8 T cells | MIS-C T1 | MIS-C T3 | 0.003 | 0.008 |
| Mitogen stimulation | Naïve effector CD4 T cells | HPC | MIS-C T1 | 0.012 | 0.036 |
| SARS-CoV-2 antigen stimulation | TNFα Naive T reg cells | MIS-C T1 | MIS-C T3 | 0.005 | 0.014 |

**Fig. 4 | Targeted analyses of immune cell data following cell stimulation assays.** Patients with multisystem inflammatory syndrome in children at timepoints 1 and 3 (MIS-C T1, $n = 8$; MIS-C T3, $n = 6$) and healthy pediatric controls (HPC, $n = 3$) were included. Heatmap of variables associated with (**a**), innate immunity and (**b**), the T cell immune response are summarized using an unsupervised hierarchical clustering model with cohort (disease group and timepoint) plotted on the x axis. Cells were measured in three states: no stimulation, stimulation with SARS-CoV-2 antigen and stimulation with mitogen. (**c**), Significant results from two-sided Wilcoxon pairwise comparisons are shown with unadjusted and adjusted (Benjamini–Hochberg) p values. Source data are provided as a Source Data file.

comparison with viral infection (Fig. 5a). One of the genes encoding FcγRIII (*FCGR3A*) was elevated in MIS-C, bacterial infection and KD in comparison with viral infection. We also analyzed genes associated with formation of NETS[34,35]: *ELANE* (neutrophil elastase) and *MPO* (myeloperoxidase) were elevated in MIS-C in comparison with viral infection (*ELANE* only) and KD (Fig. 5b). Targeted analysis of *MMP9* (matrix metallopeptidase 9), *PADI4* (peptidyl arginine deiminase 4) and *LTF* (lactotransferrin) showed these to be elevated in MIS-C and

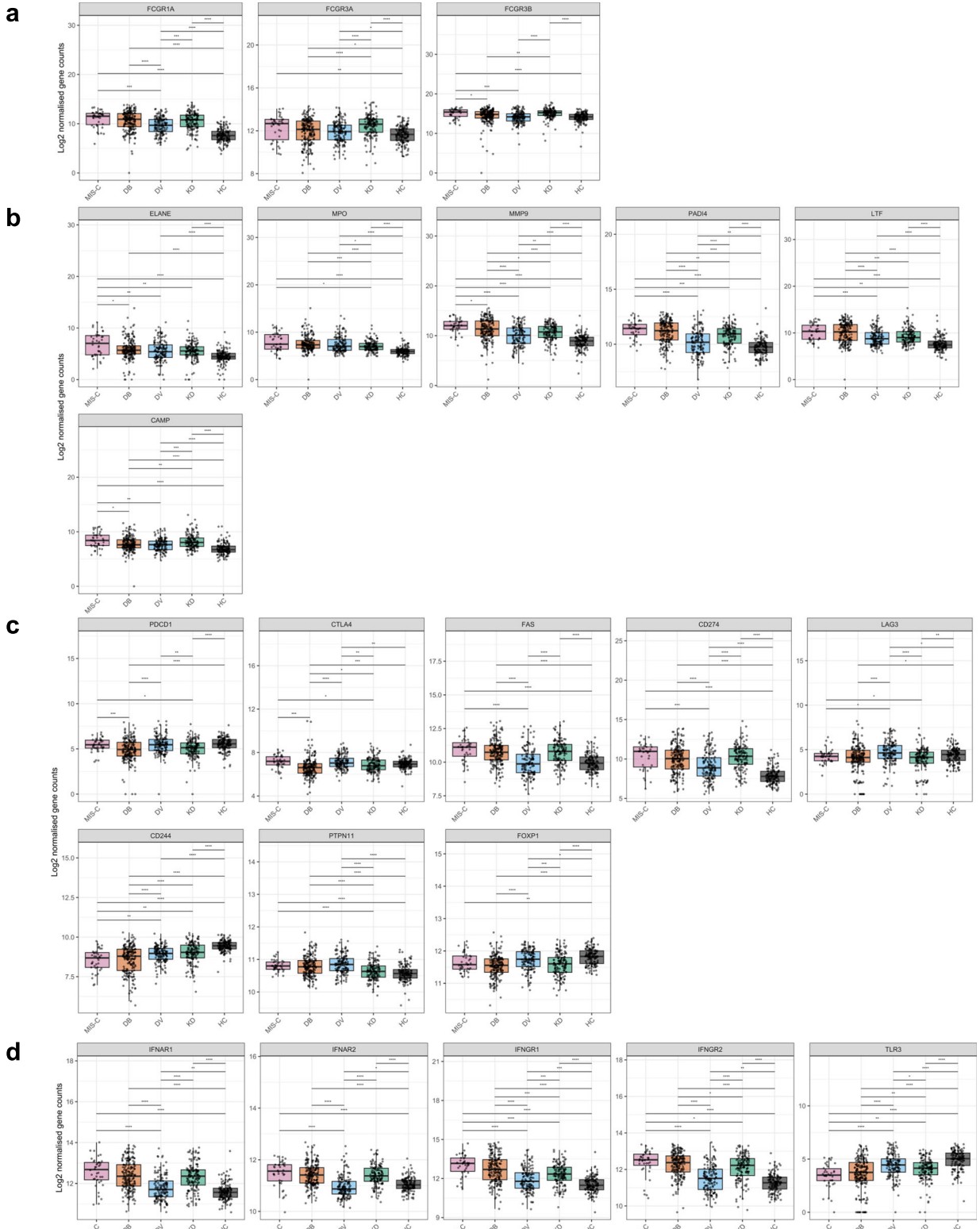

**Fig. 5 | Whole blood gene expression counts generated by RNA-Seq.** Samples are from children with MIS-C (*n* = 38), bacterial infections (DB, *n* = 188), viral infections (DV, *n* = 138), Kawasaki disease (KD, *n* = 136), and healthy controls (HC, *n* = 134). Gene counts were normalized and log₂-transformed. **a** Genes involved with formation of neutrophil extracellular traps (NETs). **b** Genes encoding FcγRI and FcγRIII. **c** Genes implicated in T cell exhaustion. **d** Genes associated with interferon signaling. Data shown are normalized to healthy pediatric controls for this cohort. Two-sided Wilcoxon pairwise comparisons with Benjamini Hochberg correction were performed; only significant *p* values are displayed. *=0.05, **=0.01, ***=0.001, ****=0.0001. Boxplots represent the median (horizontal line), first and third quartile boundary (box) and 1.5 times the inter-quartile range (whiskers) with all data points shown. Source data are provided as a Source Data file.

bacterial infection in comparison with viral infection and KD. Levels of *CAMP* (cathelicidin antimicrobial peptide) were elevated in MIS-C in comparison with bacterial and viral infections but not KD.

Regarding T cell function, expression of *PDCD1* (encoding CD279/ programmed cell death (PD)1) and *CTLA4* (CTLA4) was elevated in MIS-C in comparison with bacterial infection and KD, but not viral infection (Fig. 5c). Expression levels of both *FAS* (CD95/Fas) and *CD274* (CD274, PD-ligand(L)1) were elevated in MIS-C, bacterial infection and KD in comparison with viral infection. Contrary to the mass cytometry findings, *LAG3* (lymphocyte activating gene) expression was reduced in MIS-C and bacterial infection in comparison with viral infections. Levels of *CD244* (a pluripotent immunoreceptor) were reduced in MIS-C and bacterial infection in comparison with viral infection and KD. *PTPN11* (SHP-2, a regulator of cytokine signaling) was elevated in MIS-C, bacterial and viral infection in comparison with KD. *FOXP1* (pluripotent transcription factor, including regulation of endothelial activation) was elevated in viral infection in comparison with bacterial infection and KD but not MIS-C. Taken together, this suggests markers of immune cell exhaustion are elevated in the whole blood transcriptome in MIS-C and bacterial infection (Fig. 5), consistent with Factor 2 associations (Supplementary Figs. 6–10), in comparison with viral infection and KD.

There was an elevated expression of *IFNAR1* (IFNα/β receptor subunit 1), *IFNAR2* (IFNα/β receptor subunit 2), *IFNGR1* (IFNγ receptor subunit 1) and *IFNGR2* (IFNγ receptor subunit 2, Fig. 5d) in MIS-C, bacterial infection and KD in comparison with viral infection. We observed increased *TLR-3* (toll-like receptor 3) expression in viral infection and KD in comparison with MIS-C and bacterial infection.

## Discussion

We compared the immunology of severe febrile illnesses in children over the trajectory of illnesses[1,24,27]. This approach is supported by the framing of sepsis as a dysfunctional host response to infection[22], the successful use of immunomodulator therapies during the COVID-19 pandemic[55], and ongoing trials in sepsis[1,24]. Using mass cytometry for peripheral blood immunophenotyping, cell stimulation assays and gene expression data with supervised, and unsupervised analytic approaches, we identified several shared and distinct immunological pathways in MIS-C, SBI, SVI and KD.

In accordance with previous work, we observed highly activated neutrophils and other innate immune cells in MIS-C and SBI[14–16], and in KD. Neutrophil expression of FcγRs enables neutrophil phagocytosis in SBI. However, neutrophils from children with MIS-C had a distinct immune signature with extremely high expression of FcγRI and CD95/Fas. Upregulation of FcγR expression through IFNγ signaling may predispose neutrophil degranulation. Our gene expression analysis showed increased expression of genes associated with neutrophil activation, degranulation and production of NETs in acute MIS-C, bacterial infections and KD compared with viral infections and HPC. We did not, however, directly measure NET formation with microscopy and measurements of cell free DNA. However, other groups have shown extensive NET formation in MIS-C[32]. Formation of NETs by activated and apoptotic neutrophils may induce endothelial damage and thrombosis in MIS-C[32], COVID-19 and acute respiratory distress syndrome[36], and KD[37]. Further experiments to measure, define mechanisms and explore immunomodulation of NETosis in severe febrile illness in may be useful. Staining for intracellular cytokine production also showed neutrophils and monocytes to be producers of the myocardial depressant IL6[38], blockade of which is evidence-based in COVID-19[44]. This may be a shared pathway to myocardial dysfunction in MIS-C and SBI.

Our data support the hypothesis that T cell exhaustion is associated with MIS-C. At presentation, markers of apoptosis were increased in CD4$^+$ and CD8$^+$ T cells in comparison with other severe febrile illnesses. Unsupervised hierarchical clustering of

immune cell features showed two clusters of children: Cluster 1 was characterized by higher FCγRI expression on myeloid cells, memory T cell exhaustion and apoptosis and markers of poor antigen presentation in comparison to Cluster 2. Cluster 1 membership was associated with MIS-C and increased requirement for vasoactive infusions. Ex vivo stimulation of whole blood with SARS-CoV-2 antigen, resulted in lower IFNγ response in acute MIS-C compared to convalescent MIS-C and healthy COVID-19 vaccinated adults, highlighting the importance of antigen-specific immune cell exhaustion in MIS-C. The cell stimulation assay also showed the lack of upregulation of cytokine production (IFNγ, IL1β, IL6, IL10, IP10, TNFα, GM-CSF) in acute MIS-C, SBI, and SVI following mitogen stimulation, which was restored in convalescence. Intriguingly, cell stimulation assay results differed by Cluster membership, with children with MIS-C in Cluster 1 producing significantly more IFNγ in response to mitogen (and non-significantly to SARS-CoV-2 antigen) than those in Cluster 2, but not other cytokines, suggesting a central role of IFNγ. In the whole blood differential gene expression analysis we noted increased expression of receptors for IFNα/β and for IFNγ in children with MIS-C, bacterial illness and KD in comparison with viral illness, implying continued upregulation of inflammatory responses in these illnesses. It is unclear if the relative down-regulation of *IFNAR1/IFNAR2/IFNGR1/IFNGR2* in SVI is a physiological response to limit over-exuberant inflammation or is a pathological process. Conversely, *TLR3* is required for clearance of herpes simplex virus[39], but implicated its upregulation is implicated in damaging innate and adaptive immune responses in respiratory infection[40]. Future work should explore the association of IFNγ signaling with T cell exhaustion in severe febrile illnesses in children.

We deliberately focused on sampling children with severe illness since these children are the most likely to be treated with immunomodulator therapies in the acute setting. Our work was limited by the availability of samples from children with KD and other rare inflammatory diseases, and the limited number of samples in mass cytometry experiments following cell stimulation assays. In the cell stimulation assays, stimulation with a toll-like receptor agonist may have informed the functional innate immune response in MIS-C and HPC. We did not examine Vβ expansion in this study, nor precipitate spike protein-immunoglobulin complexes, limiting our analysis of these important topics. Of children with samples for which mass cytometry was acquired, half of children with MIS-C had been treated with a first dose of corticosteroid or a first dose of immunoglobulins. However, a sensitivity analysis showed no association with treatment and immune features. A key limitation to many observational studies of childhood disease is the lack of sampling prior to illness. This limits our ability to infer differences between causal pathways to disease, and "bystander" inflammation. Further ex vivo experiments may be useful, particularly regarding modulation of neutrophil activation.

In summary, we used a multi-omic systems biology approach to provide a comparative analysis of the trajectory immunology of severe febrile illness in children in the COVID-19 era. Our data support an important role for neutrophil and monocyte activation in the pathology of MIS-C and highlight T cell exhaustion upon presentation with MIS-C. Neutrophil activation and features of T cell activation and exhaustion were shared with SBI, while SVI was characterized by downregulated STAT signaling pathways, highlighting shared and distinct features of immune dysregulation in these disparate severe febrile illnesses of childhood.

## Methods
### Settings and approvals
Patients and controls for this study were recruited at Evelina London Children's Hospital, St Mary's Hospital London, Queen Elizabeth

Hospital, Woolwich, London and Lewisham Hospital, London between July 2020 and May 2022. The current cohort study was approved by the UK HRA (20/HRA/1714; for further details please see https://www.diamonds2020.eu/about/). Informed consent was obtained (from parents or guardians of children <16 years of age) by trained health professionals. The STROBE (Strengthening the Reporting of Observational Studies in Epidemiology) checklist compliance is provided in the Supplementary Table 7.

## Clinical sampling for cytometry and cell stimulation cohort
Eligibility criteria were suspected infection or inflammatory disease and weight ≥ 2.5 kg. Patients admitted to the Pediatric Intensive Care Unit were prioritized for recruitment. Samples from children in the cohort were selected for analysis if they were obtained from children acutely unwell with either MIS-C (as per WHO definition, the preferred definition at the time)[28], COVID-19 pneumonitis (as per WHO definition)[41], clinically diagnosed KD (as per American heart association guidelines[29], with or without SARS-CoV-2 IgG antibody), SVI, SBI, or a clinically diagnosed inflammatory disease. Sex was not included in the study design and was self/parental-reported as sex at birth. Ethnicity was self-reported by participating families on case record forms.

Sampling time points were acute (as soon as practicable on admission to hospital, and using deferred consent), defervescence (no fever for ≥24 h and serum CRP < 100 mg/l) and convalescence (clinical review following discharge). Research samples included whole blood for immunophenotyping and cell stimulation assays, and, where available, plasma for cytokine analysis.

## Clinical sampling for the transcriptomic cohort
Patients were recruited to the cohort as described in Jackson et al.[27]. In brief, patients with suspected severe infection or severe inflammatory disease were recruited to ongoing successive European Union-funded international studies: DIAMONDS (Diagnosis and Management of Febrile Illness using RNA Personalised Molecular Signature Diagnosis) and PERFORM (Personalised Risk assessment in Febrile illness to Optimise Real-life Management across the European Union). Venous blood was collected from patients at the earliest time point after recruitment and prior to treatment with IVIG or immunomodulator into PAXgene Blood RNA (PreAnalytiX, Germany) and stored at −80 °C until RNA extraction. Patients recruited to each study were phenotyped according to our published algorithm[31], and as used to develop whole blood transcriptional signatures of severe febrile illnesses in children[29]. KD was diagnosed using the American Heart Association diagnostic criteria[30], and MIS-C was diagnosed based on the WHO criteria[28]. We included patients with MIS-C, KD, confirmed bacterial infection (termed definite bacterial; DB), and confirmed viral infections (definite viral; DV). Healthy control children were included but only used to correct for batch effects between groups. DV was conditional upon the identification of a virus compatible with the clinical syndrome with no evidence of bacterial infection and CRP ≤ 60 mg/L.

## Mass cytometry
**Samples directly from patients.** Samples of 500 μl of whole blood from patients were pipetted directly into 500 μl of Cytodelics cell stabilizing solution (Cytodelics ab, Sweden) and stored within 30 min at −80 °C. Following storage, samples were initially processed by rapid thawing in a water bath at 37 °C for 1–2 min, fixing in 2500 μl Cytodelics fixation buffer with the addition of 50 μg EDTA (250 μl of 0.5 mM/ 0.02% solution; Thermofisher, MA, USA) at 20 °C for 15 min before lysis of red cells using 5 ml Cytodelics lysis buffer for 20 min. Samples were subsequently washed (10 ml Cytodelics wash buffer) twice with centrifugation set at 1000 g for 5 min at 20 °C, before transfer from a 15 ml Falcon tube to a microcentrifuge tube for further processing. Unless otherwise stated, all further centrifugation steps were at 1000 g for 5 min at 20 °C. Cell staining buffer (CSC), Fix and Perm Buffer (FPB) and

Cell Acquisition Solution (CAS) are all Fluidigm Inc. (CA, USA, now Standard Biotools). A single standardized healthy control sample was stained and analyzed for each batch of samples.

Cells were centrifuged and then resuspended in 50 μl Fc receptor blocking solution (45 μl cell staining buffer (CSB)/5 U heparin/5 μl FcX Trustain, Biolegend, CA, USA) and incubated for 10 min at 20 °C. *Barcoding antibody mastermixes* to CD45 (Fluidigm, now Standard Bio-Tools; Supplementary Fig. 2) were then added to individual samples to allow later deconvolution of data and incubated for 20 min at 20 °C with gentle mixing. Cells were washed twice in CSB and groups of samples were created by pooling six predefined samples into a single tube for subsequent multiplexing. Cells (now six-plex in each tube) were washed and resuspended in 50 μl Fc receptor blocking solution and incubated for 10 min at 20 °C, before washing twice in CSB. Cells were then resuspended in *cell surface antibody mastermix* and incubated for 30 min at 20 °C with gentle mixing. Cells were then washed in CSB twice before being resuspended in 1000 μl FPB and incubated for 10 min at 20 °C, before being washed in FPB twice. Cells were then resuspended in *intracellular cytokine mastermix* and incubated for 30 min at 20 °C with gentle mixing before being washed twice in CSB and placed on ice for 10 min. 700 μl of neat methanol (taken directly from a -20 °C freezer) was then added dropwise to tubes of cells before incubation for 15 min on ice. 700 μl cold CSB (placed on ice) was then added to each tube and tubes were centrifuged at 1500 g at 4 °C for 5 min, with supernatant discarded; a further 1000 μl of cold CSB was added to each tube and centrifugation at 1500 g at 4 °C was repeated before aspiration of supernatant. Cells were then resuspended in *phosphorylated mastermix* and incubated for 45 min at 4 °C. Cells were then washed in CSB twice and resuspended in 1000 μl of intercalator solution (Cell-ID intercalator-Ir193 125 μM diluted to 0.125 μM in phosphate buffered saline with 2% paraformaldehyde made freshly). Cells were incubated for 30 min at 20 °C, then centrifuged at 1200 g and 20 °C for 5 min, with most supernatant aspirated and cells resuspended in -100 μl remnant intercalator solution and frozen immediately at −80 °C. Stained six-plex tubes of cells were subsequently thawed at 37 °C, washed in intercalator solution, and then incubated for 60 min at 20 °C. Cells were then washed in CSB twice, and then CAS twice, resuspended in CAS to an appropriate concentration before addition of EQ beads (Fluidigm). Data was then immediately acquired using a Fluidigm Helios Mass Cytometer at the Advanced Cytometry Platform (Flow Core), Research and Development Department at Guy's and St Thomas' NHS Foundation Trust (https://sites.google.com/nihr.ac.uk/brcflow/home).

Samples were acquired in cell acquisition solution (CAS) with 1:10 EQ™ four element calibration beads using a Helios mass cytometer (Fluidigm). Flow cytometry standard (FCS) files were then normalized with EQ beads for comparison between samples. Files were then concatenated using the CYTOF software (Fluidigm). We used CD45 labels ("barcodes") to enable pooling of six-plexed samples and subsequently debarcoding of samples from FCS files using a custom MATLAB script and manual gating on FlowJo based on positive then negative selection by CD45 barcodes (BD, version 10.8.1). FCS files of debarcoded samples were then imported into CytofClean (https://rdrr.io/github/JimboMahoney/cytofclean/man/cytofclean.html) on R[42] for further clean up and auto-gating based on the Gaussian parameters (Event Length, Center, Offset, Residual and Width). To normalize and correct for intra-experiment variability a whole blood technical control was run together with all the batches in each experiment and used for normalization using the FlowJo plugin Cytonorm. Samples were then imported on the online platform Cytobank for further informatic analysis (https://www.cytobank.org).

**Samples from cell stimulation assays.** Samples of 100 μl of blood from cell stimulation assays were used. Following storage, samples were initially processed by rapid thawing in a water bath at 37 °C for

1–2 min, fixing in 2500 μl Cytodelics fixation buffer with the addition of 50 μg EDTA (250 μl of 0.5 mM / 0.02% solution; Thermofisher, MA, USA) at 20 °C for 15 min before lysis of red cells using 5 ml Cytodelics lysis buffer for 20 min. Samples were subsequently washed (10 ml Cytodelics wash buffer) twice with centrifugation set at 1000 g for 5 min at 20 °C, before transfer from a 15 ml Falcon tube to a microcentrifuge tube for further processing. Unless otherwise stated, all further centrifugation steps were at 1000 g for 5 min at 20 °C. Cell staining buffer (CSC), Fix and Perm Buffer (FPB) and Cell Acquisition Solution (CAS) are all Fluidigm Inc. (CA, USA, now Standard Biotools).

Cells were centrifuged and then resuspended in 50 μl Fc receptor blocking solution (45 μl cell staining buffer (CSB)/5 U heparin/5 μl FcX Trustain, Biolegend, CA, USA) and incubated for 10 min at 20 °C. *Barcoding antibody mastermixes* to CD45 (Fluidigm, now Standard Bio-Tools; Supplementary Fig. 2) were then added to individual samples to allow later deconvolution of data and incubated for 20 min at 20 °C with gentle mixing. Cells were washed twice in CSB and groups of samples were created by pooling six predefined samples into a single tube for subsequent multiplexing. Cells (now six-plex in each tube) were washed and resuspended in 50 μl Fc receptor-blocking solution and incubated for 10 min at 20 °C, before washing twice in CSB. Cells were then resuspended in *cell surface antibody mastermix* and incubated for 30 min at 20 °C with gentle mixing. Cells were then washed in CSB twice before being resuspended in 1000 μl FPB and incubated for 10 min at 20 °C, before being washed in FPB twice. Cells were then resuspended in *intracellular cytokine mastermix* and incubated for 30 min at 20 °C with gentle mixing before being washed twice in CSB twice and resuspended in 1000 μl of intercalator solution (Cell-ID intercalator-Ir193 125 μM diluted to 0.125 μM in phosphate-buffered saline with 2% paraformaldehyde made freshly). Cells were incubated for 30 min at 20 °C, then centrifuged at 1200 g and 20 °C for 5 min, with most supernatant aspirated and cells resuspended in ~100 μl remnant intercalator solution and frozen immediately at −80 °C. Stained six-plex tubes of cells were subsequently thawed at 37 °C, washed in intercalator solution, and then incubated for 60 min at 20 °C. Cells were then washed in CSB twice, and then CAS twice, resuspended in CAS to an appropriate concentration before the addition of EQ beads (Fluidigm). Data was then immediately acquired using a Fluidigm Helios Mass Cytometer at the Advanced Cytometry Platform (Flow Core), Research and Development Department at Guy's and St Thomas' NHS Foundation Trust (https://sites.google.com/nihr.ac.uk/brcflow/home).

**Ex vivo cell stimulation assays and supernatant cytokine analysis.** 4 ml of whole blood was sampled directly from patients recruited to the study and with a body weight ≥20 kg into a lithium heparinized tube (Vacutainer, Becton Dickinson, MA, USA) and transported directly to the laboratory (working hours) or stored overnight at 4 °C before transport. On receipt in the laboratory, 1000 μl of heparinized blood was added to each of 4 QuantiFERON SARS-CoV-2 RUO assay tubes, containing either SARS-CoV-2 RBD spike protein peptide ("Ag 1") to stimulate cognate CD4+ T cells, blank (negative control; "nil") and mitogen (phytohemagglutinin, positive control, "mit") (Qiagen, Netherlands). QuantiFERON SARS-CoV-2 tubes were incubated at 37 °C for 17–24 hours. Following incubation, the entire blood volume of each tube was placed into a 5 ml screw top tube, gently mixed and 100 μl of blood was aspirated and pipetted into a Cryovial containing 100 μl Cytodelics cell stabilization solution, incubated at room temperature for 10 min and transferred to −80 °C storage. The remaining volume was then centrifuged at 3000 g for 15 min, before removal and storage of supernatant at −80 °C. Supernatant from the stimulated samples was separated and used for detection of cytokines by cytometric bead arrays using the Legendplex human anti-virus response panel (BioLegend, #740390) to measure cytokines levels of IL1β, TNFα, IL6, IL8, IL10, IL12p70, IP10, GM-CSF, IFNα2, IFNβ, IFNγ, IFNλ1, and IFNλ2/3. The

kits were used according to the manufacturer's instructions and samples tested in duplicate.

**Supernatant cytokine data acquisition and analysis.** A BD LSR Fortessa flow cytometer was used for data acquisition with PE as the reporter channel (575–585 nm) and APC used as the classification channel (660 nm). Compensation was not required. The LEGENDplex (Biolegend) guide was followed to set up the PMT voltages with the set-up beads using the FACSDiva software (Version 6.0). The HTS was used for rapid automated sample acquisition and the total events per well were limited to 10,000.

The LEGENDplex Data Analysis Software Suite (Version 8) was used for initial quality control checks. The automated gating of bead population was reviewed and manually adjusted for any misclassifications. A 5-parameter logistic (PL) equation was used to fit the standard curves using logistic regression. The threshold for the $R^2$ value was set at >0.95. A minimum of 100 events were required for each bead ID and the threshold for the coefficient of variation between replicates was set at <30%. Principle component analyses (PCA) were performed to detect any batch effects as the samples were run across 20 plates. Raw data were used for baseline, unstimulated samples. Percentage change from baseline was used for evaluating the stimulated samples. To address the skewness of data and to improve visualization of data, the results were presented after applying log10 transformation. Pairwise comparisons using the Wilcoxon test were performed across clinical cohorts and timepoints with BH[44] correction for multiple testing. A sub-analysis of the MIS-C T1 (acute) cohort by the two clusters identified from the unsupervised hierarchical clustering analysis of the mass cytometry data was performed. Wilcoxon pairwise comparisons were performed to assess differences in cytokine levels between Clusters 1 and 2. Significance level was set at $p < 0.05$.

### Whole blood RNA Sequencing

The whole blood RNA Sequencing (RNA-Seq) data used in this study has been described previously by Jackson et al.[27], where the full details of the sequencing protocol are described in the Supplementary Methods.

Total RNA was isolated using recommended methodology (including miRNA for PAXgene blood RNA vacutainers and after additional DNAse treatment and sent for RNA-Seq at The Wellcome Center for Human Genetics in Oxford, United Kingdom using a Novaseq6000 platform at 150 bp paired-end configuration, generating a raw read count of 30 million reads per sample. Raw reads were aligned to the human genome (build GRCh38, available from http://www.gencodegenes.org/releases/21.html). Following alignment and filtering, we used the software HTSeq[45] to generate counts of the number of filtered reads mapped to each gene isoform within the human genome build).

We used the lists of significantly differentially expressed (SDE) genes described in Jackson et al.[27] to determine whether specific genes of interest were SDE between MIS-C and any of the other illness groups (KD, DB, DV). Lists of SDE genes were identified by Jackson et al. using the negative binomial distribution implemented in DESeq2 (https://bioconductor.org/packages/release/bioc/html/DESeq2.html). Genes with a BH adjusted *p*-value < 0.05 were considered SDE. Furthermore, targeted statistical tests using the Wilcoxon test were used to compare levels of the genes between groups. This further step was performed in case some genes did not meet the stringent multiple testing step from differential expression analysis. All statistical analyses were performed using the statistical software R (R version 4.0.3).

Gene-set enrichment analysis (GSEA)[33] (Supplementary Fig. 16) was undertaken on previously published data from Jackson et al.[29] that are available at ArrayExpress under accession E-MTAB-11671 and E-MTAB-12793. We used the R (version 4.4) implementation of gene-set

enrichment analysis (*fgsea*, version 1.3)[41], the C7 immunologic signature gene sets available from as part of the Human MSigDB Collections (https://www.gsea-msigdb.org/gsea/msigdb/collections.jsp), and Gene Ontology (GO) enrichment analysis. We compared gene enrichment between children with MIS-C and healthy controls, with definite bacterial infection and healthy controls, definite viral infection and healthy controls and children with MIS-C and definite bacterial infection. Code for the GSEA is available here: https://github.com/michaeljamescarter/SIFIC.

#### Informatic analysis of mass cytometry data
**Barcoding and adjusting to healthy controls.** Identification of populations identified by manual gating enabled analysis of immune cell population proportions (proportions) and cell activation state by surface protein, intracellular cytokine expression, and phosphorylation (p) status of signal transduction and activator of transcription (STAT)1, STAT5, nuclear-factor-κ-B (NFκB), and S6 in the mammalian target of rapamycin, mTOR, pathway) (cell population expression markers).

**Supervised analysis.** We used supervised manual gating to identify canonical cell populations using Cytobank (v 10.3, Beckmann Coulter, USA). Single-cell data were down sampled to a total of 100,000 cells per sample. Supervised analysis was done in Cytobank using the gating strategy described in Supplementary Fig. 3 to describe the proportions of major cell populations within the sample (Supplementary Fig. 3). Median expression levels of cell state markers (either cell surface markers, intracellular cytokines or phosphorylated signaling pathways) were then extracted from Cytobank to form cell population expression data. Additionally, each manually gated population for each FCS file was extracted by Cytobank and combined to create heatmaps for each group of patients with severe febrile illness at each time point.

**Normalization of data to healthy pediatric controls.** Analysis of healthy pediatric controls (HPC) showed low variance of all markers for gating of cell population proportions and for cell expression markers (see Fig. 2a–c), except HLA-DR expression on plasmablasts, CD38 expression on plasmablasts, CD27 expression on plasmablasts and HLA-DR on plasmacytoid DCs. These four markers with high variance were subsequently deleted from all analyses. We subsequently took the median value of cell population proportions and cell population expression markers and used these values to scale data from individual children with severe febrile illness. The scaled data were then used for subsequent analyses, including general linear models, hierarchical clustering, and multi-omic factor analysis (MOFA). Data from children with severe febrile illnesses expressed in these analyses therefore represent the "distance" from HPC data.

**General linear models.** We iteratively undertook general linear modeling for associations between any of the immune features (cell population proportions or cell expression markers) and disease group (MIS-C, SBI, SVI or KD) at T1 using data scaled to HPCs (as described above). No covariates were controlled for. We used the Wilcoxon Test with BH correction for multiple testing to assess the significance of associations. Immune features that had a significant association ($p < 0.050$ following BH correction) are described in Fig. 2b.

**Hierarchical clustering.** We took markers that were differentially expressed in the GLM (all at T1) and entered these markers into hierarchical clustering algorithms. We used sum of squares, silhouette and the Gap statistic to determine two clusters as the most parsimonious number of clusters and Ward's method of hierarchical clustering as implemented by the R packages stats[42] and *pheatmap* (https://www.rdocumentation.org/packages/pheatmap/versions/1.0.12/topics/

pheatmap). We subsequently termed the clusters Cluster 1 and Cluster2 and described the clinical features and immune features associated with cluster membership.

For mass cytometry of blood following cell stimulation assays, data were not normalized to healthy pediatric controls. Hierarchical clustering of the innate and T cell immune response (Figs. 4a and 4b) was done using Ward's method as implemented by *pheatmap*.

**Multi-omic factor analysis.** Rational dimensionality reduction methods are necessary when interpreting high-parameter datasets to limit spurious comparisons and resulting bias. We used MOFA[28] to investigate the association of cell population proportions and cell population expression data with clinically relevant variables for illness classification, time point, treatment, and severity. MOFA is an unsupervised computational method to discover the principal axes of variability within high parameter datasets. The resulting factors can be interpreted like a principal component analysis.

#### Mass cytometry analysis of stimulated samples
Raw values were used for unstimulated comparisons: unstimulated MIS-C T1 vs unstimulated healthy pediatric controls, unstimulated MIS-C T3 vs unstimulated healthy pediatric controls. The median results for each variable, by cohort, were represented on heatmaps with hierarchical clustering. To identify significant features within cell proportions and cell expression markers data, multiple Wilcoxon pairwise comparisons with Benjamini-Hochberg adjustment (threshold $p < 0.05$) were performed. To assess the effect of stimulation and to bring the data onto the same scale, fold change from unstimulated data to stimulated data were calculated for each individual. The median fold change for each variable, by cohort, were represented on heatmaps with hierarchical clustering. Statistical analyses were performed using the statistical software R (R version 4.2.1).

#### Ethics statement
The Derivation cohort were recruited to the observational study, DIAMONDS Search Study (Diagnosis and Management of Febrile Illness using RNA Personalised Molecular Signature Diagnosis; https://www.diamonds2020.eu) under UK HRA approval 20/HRA/1714 following informed written parental consent (children under 16 years of age) or informed written consent with parental assent (young people 16 to 18 years of age). The IRAS ID is 278651. The Transcriptomic cohort were recruited to the observational studies EUCLIDS (European Union Childhood Life-Threatening Infectious Disease Study, UK HRA approval 16/LO/1684), PERFORM (Personalised Risk Assessment in Febrile Illness to Optimise Real-Life Management across the European Union, UK HRA approval 11/LO/1982) and the DIAMONDS Search Study. Participants were not compensated.

#### Inclusion and ethics
This study was designed and implemented by clinicians and researchers embedded within treating clinical teams and this is reflected in the opportunities to contribute meaningfully to the data analysis, interpretation, and writing. This is reflected in the authorship. Members of the wider DIAMONDS Search Study consortium are identified in the Supplementary Information.

#### Reporting summary
Further information on research design is available in the Nature Portfolio Reporting Summary linked to this article.

## Data availability
The gene counts and patient metadata for the Transcriptomic Cohort are available at ArrayExpress under accession code E-MTAB-11671. The

merged and normalized dataset used for the analysis of the discovery dataset is available in ArrayExpress under accession code E-MTAB-12793. The raw FCS files for Supplementary Fig. 10 are available on ImmPort under accession code SDY2735. Source data are provided with this paper.

## Code availability

Code for the bioinformatic analyses of mass cytometry data is available at: https://github.com/michaeljamescarter/SIFIC. Code for the analysis of RNA-seq data is available at: https://github.com/PIDBG/misc_transcriptomic_signature[29]. A permanent repository for the code is at the https://doi.org/10.5281/zenodo.12790924.

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

## Acknowledgements

This study was supported by a National Institute for Health Research (NIHR) Academic Clinical Lectureship (NIHR-ACL-2018, MJC), by a King's Together grant from King's College London, by funding from the Huo Family Foundation (M.J.C., M.S.H. and M.H.M.), by the Evelina London Children's Hospital PICU Research Fund (S.M.T.). This work was also supported by a NIHR Clinician Scientist Fellowship (2016-16-011, M.S.H.) and laboratory support from the Chief Scientists Office, Scotland, for time- critical precision medicine in adult critically ill patients and highlights industry interactions for TRAITS research program (https://www.ed.ac.uk/inflammation-research/clinical-trials/traits-ci-trial) (MSH) and by Research and Development, Guy's and St Thomas' NHS Foundation Trust (R.E., M.T.B., F.S., N.P., C.B., K.T.). The views expressed are those of the author(s) and not necessarily those of the NHS. This work was supported by the European Union's Horizon 2020 Program under grants (848196 DIAMONDS, 668303 PERFORM, 279185 EUCLIDS, M.L.), by the Imperial Biomedical Research Centre (BRC) of the NIHR, Wellcome Trust and the Medical Research Foundation grants (206508/Z/17/Z and MRF-160-0008-ELP-KAFO-C0801, M.K.; 215214/Z/19/Z to H.J.) and grant from the Community Jameel Imperial College COVID-19 Excellence Fund and the Rosetrees Trust (WDPI_G28062 to J.H.). This work has been supported by the Imperial Confidence in Concept Scheme, funded by MRC Confidence in Concept, Wellcome Trust Institutional Strategic Support Fund, NIHR Imperial BRC, and Rosetrees Trust (MK). QuantiFERON SARS CoV-2 RUO tests were provided by Qiagen. Qiagen had no role in the study design, data analysis and manuscript preparation. We acknowledge the support of the DIAMONDS Consortium (https://www.diamonds2020.eu/). A full list of DIAMONDS Consortium members can be found in the Supplementary Information.

## Author contributions

H.P., M.J.C., P.W., S.D., S.M., J.M.C., J.K., S.v.d.B., T.S., F.D., J.T., M.N., T.D., E.T., P.R., E.W., J.H., A.C., S.M.T., and M.L. recruited patients and collected samples and analyzed clinical data. M.J.C., M.F., M.S.H., K.T., C.B., S.v.d.B., H.P., A.J., R.E., M.H.M. designed and undertook mass cytometry experiments. H.P., O.P., P.W., M.J.C., M.L. designed and undertook cell stimulation experiments. H.J., S.N., L.E.E., G.D.S., M.J.C., V.W., D.H.C., J.H., A.C., M.K., and M.L. designed and undertook gene expression experiments. M.J.C., M.T.B., F.S., T.S., N.P., P.T., K.T., C.B., and R.E. undertook gating and bioinformatic analysis of mass cytometry data. Dr Carter and Dr Patel contributed jointly to the work as joint first authors. Dr Jackson and Mr Powell contributed jointly to the work as joint second authors. Prof Shankar-Hari and Prof Levin contributed jointly to the work as supervisors of the work and corresponding authors. All authors contributed to the writing and review of the manuscript and have approved the submitted and revised versions.

## Competing interests

The authors declare no competing interests.

## Additional information

Harsita Patel[1,12], Michael J. Carter [2,3,12], Heather Jackson [1], Oliver Powell[1], Matthew Fish[4], Manuela Terranova-Barberio[5,6], Filomena Spada[5], Nedyalko Petrov [5], Paul Wellman [3], Sarah Darnell[1], Sobia Mustafa[1], Katrina Todd[5], Cynthia Bishop[5], Jonathan M. Cohen [7], Julia Kenny[7], Sarah van den Berg [3], Thomas Sun[3], Francesca Davis [7], Aislinn Jennings[2], Emma Timms[4], Jessica Thomas[8], Maggie Nyirendra[8], Samuel Nichols [1], Leire Estamiana Elorieta[1], Giselle D'Souza[1], Victoria Wright [1], Tisham De [1], Dominic Habgood-Coote [1], Padmanabhan Ramnarayan[9], Pierre Tissières [10], Elizabeth Whittaker[1], Jethro Herberg[1], Aubrey Cunnington [1], Myrsini Kaforou [1], Richard Ellis [5], Michael H. Malim [4], Shane M. Tibby[2,3], Manu Shankar-Hari [11,12] ✉, Michael Levin [1,12] ✉, On behalf of the DIAMONDS Consortium

[1]Section of Infectious Diseases, Department of Medicine, St Mary's Hospital Campus, Imperial College London, Praed Street, London, UK. [2]Department of Women and Children's Health, School of Life Course and Population Sciences, King's College London, St Thomas' Hospital, Westminster Bridge Road, London, UK. [3]Paediatric Intensive Care, Evelina London Children's Hospital, Guy's and St Thomas' NHS Foundation Trust, Westminster Bridge Road, London, UK. [4]School of Immunology and Microbial Sciences, King's College London, Guy's Hospital, Great Maze Pond, London, UK. [5]Advanced Cytometry Platform (Flow Core), Research and Development Department at Guy's and St Thomas' NHS Foundation Trust, Guy's Hospital, Great Maze Pond, London, UK.

[6]Flow Cytometry Core, Barts Cancer Centre, Queen Mary University of London, John Vane Science Centre, Charterhouse Square, London, UK. [7]Paediatric Immunology and Infectious Diseases, Evelina London Children's Hospital, Westminster Bridge Road, London, UK. [8]Children's Services, Lewisham and Greenwich NHS Foundation Trust, London, UK. [9]Department of Surgery and Cancer, St Mary's Hospital Campus, Imperial College London, London, UK. [10]Institut de la Biologie de la cellule, Université Paris Saclay, Gif-sur-Yvette, Departement de l'Essone, Gif-sur-Yvette, France. [11]Institute for Regeneration and Repair, Centre for Inflammation Research, University of Edinburgh, Edinburgh Royal Infirmary, Little France Crescent, Edinburgh, UK. [12]These authors contributed equally: Harsita Patel, Michael J. Carter, Manu Shankar-Hari, Michael Levin. ✉e-mail: manu.shankar-hari@ed.ac.uk; m.levin@imperial.ac.uk

## the DIAMONDS Consortium

**Harsita Patel**[1,12], **Michael J. Carter** [2,3,12], **Heather Jackson** [1], **Oliver Powell**[1], **Matthew Fish**[4], **Manuela  Terranova-Barberio**[5,6], **Filomena Spada**[5], **Nedyalko Petrov** [5], **Paul Wellman** [3], **Sarah Darnell**[1], **Sobia Mustafa**[1], **Katrina Todd**[5], **Cynthia Bishop**[5], **Jonathan M. Cohen** [7], **Julia Kenny**[7], **Sarah van den Berg**[3], **Thomas Sun**[3], **Francesca Davis** [7], **Aislinn Jennings**[2], **Emma Timms**[4], **Jessica Thomas**[8], **Maggie Nyirendra**[8], **Samuel Nichols** [1], **Leire Estamiana Elorieta**[1], **Giselle D'Souza**[1], **Victoria Wright** [1], **Tisham De** [1], **Dominic Habgood-Coote** [1], **Padmanabhan Ramnarayan**[9], **Pierre Tissières** [10], **Elizabeth Whittaker**[1], **Jethro Herberg**[1], **Aubrey Cunnington** [1], **Myrsini Kaforou** [1], **Richard Ellis** [5], **Michael H. Malim** [4], **Shane M. Tibby**[2,3], **Manu Shankar-Hari** [11,12]✉ & **Michael Levin** [1,12]✉

A list of members and their affiliations appears in the Supplementary Information.

