## [Peer Review File · Nature Communications]

Shared neutrophil and T cell dysfunction is accompanied by a distinct interferon signature during severe febrile illnesses in childrenREVIEWER COMMENTS

Reviewer #1 (Remarks to the Author):

Major comments:

1. In their analysis of gene expression, the authors attribute a limited number of changes in mass cytometry to highly specific cellular states. However, these few proteins are not sufficiently specific for the authors' claims:

- For example, they use CD279 as a marker for exhausted T cells, but this gene is also upregulated on T cells in response to Type II IFNs, which does not necessarily imply exhaustion. Additionally, CD279 is upregulated on recently activated T cells and on specific subsets of T cells (eg CD4+ T follicular helper cells). Similar findings and the limitations of using CD279 as a sole marker of exhaustion have been noted in patients with MIS-C (Vella, L. et al. *Sci Immunol.*, 2021).

- The authors state that "Factor 2 represented restoration of innate and adaptive immune cell homeostasis, with a decrease in the proportion of immature neutrophils and increased CD11b and CD16 on myeloid cells" (lines 218 – 220). Additionally biologic proof is needed to support these statements, as these two markers are not sufficient to indicate "restoration of innate and adaptive immune cell homeostasis"

- Similarly, the author note upregulation of "genes associated with formation of NETs^{31,32}: ELANE (neutrophil elastase) and MPO (myeloperoxidase) were elevated in MIS-C in comparison with viral infection (ELANE only) and KD (Figure 5b)" (lines 341-343). Similarly, the discussion, they state: Our gene expression analysis showed increased expression of markers related to NETs in acute MIS-C, bacterial infections and KD compared with viral infections and HPC (lines 390 – 391). These genes are upregulated with neutrophil activation and are not necessarily markers of NETs. To support this statement, multiple assays (eg bisulfite sequencing of cell-free DNA to determine cell origins, quantification of intact nucleosomes containing histone 3.1 , or citrullinated histone H3R8, among others) would need to be done.

2. For the T cell stimulation assays, the percentages of antigen-specific T cells (eg through an AIM assay) are important for understanding the data, as this may vary greatly among the cohorts and may contribute assessments of differences in cytokine production. Intracellular staining for key cytokines (eg IFN-g) in SARS-CoV-2-specific T cells would be more convincing evidence of the attenuated response to spike antigen that the authors report. These data are also complicated by the fact that for some of the patients with MIS-C, there was little response to mitogen stimulation (which is typically used as a positive control to indicate overall cell function despite a lack of specific Ag response).

3. The manuscript describes findings, but unfortunately cannot identify underlying mechanisms.

a. For example, the authors state that: "Taken together, MOFA showed differences in myeloid cell and T cell activation markers predominate differences between groups of patients with severe febrile illnesses and innate and adaptive immune cell function is restored by convalescence in MIS-

C and SBI.” (lines 227 – 230). What are the mechanisms underlying these differences?

b. Similarly, the differences in gene expression are described without additional context or follow-up investigations, as line lines 366 – 370 presented increased expression of IFNAR1, IFNGR1/2 in patients with MIS-C, SBI, and KD compared to those with SVI) and increased TLR3 in patients with SVI and KD compared to MIS-C and SBI. What is the significance of these findings?

4. This claim needs additional clarification: “Further experiments to define the mechanism of NET production in severe febrile illness in children may be useful, particularly since polyspecific intravenous immunoglobulins are used as treatment in MIS-C, KD, and toxic shock syndrome” (lines 392 – 394). Are the authors implying that IVIG influences NETs?

5. Additional support is needed to support the authors’ statement that: “More generally, this characterization suggests a key signaling role for neutrophils.” Additional studies are needed to prove that neutrophils have a key signaling role in the pathogenesis of MIS-C. How do the authors differentiate between bystander activation/apoptosis of neutrophils from a driver effect of neutrophil activation?

6. Please justify why for the human adult controls all were vaccinated but for human pediatric controls almost none were vaccinated. It is unclear to me why the healthy adult controls (HAC) were chosen as the rest of the cohorts are pediatric. It is not surprising that there are differences immunologically between non-ill vaccinated adults and severely ill febrile children. Some of the differences were between MIS-C and HAC only and not with healthy pediatric controls (HPC) eg. lines 280-283. So if HPC and MIS-C are different than HAC, would it infer that it is an age-related difference given the immune systems of adults are different than children?

7. There are no details given in the table of patients about the HAC but the age differences in the pediatric cohorts is large. For example, KD median age is 1.4 years and there are only 8 KD patients. SVI 2.9 years median age. Whereas MIS-C, SBI and other inflammatory conditions median age is 9-10 years. Age alone may explain differences.

8. The authors do not note that MIS-C overlaps with KD and many MIS-C patients meet KD criteria. Sub-phenotypes within MIS-C are also important to evaluate.

9. The authors repeatedly mention "sepsis" and note a paper by Menon showing overlap of critical illness phenotypes. The relationship of the authors work to reference is not clear to me. MIS-C is a post-infectious inflammatory process that looks like sepsis. The patients with SBI likely many have sepsis and some of the SVI. The paper by Menon compared TBI, sepsis, ARDS and other critical illness phenotypes and showed overlap of these phenotypes. The authors are trying to show differences and are not emphasizing similarities.

Minor comments

1. In the introduction, you state with certainty why the incidence of MIS-C has decreased. But it is not certain, and the authors may want to insert the word "likely" before the word "because".

2. Expressing the cytokine concentration in pg/mL or ng/mL is standard convention. The baseline units in Supplemental Table 5 are not defined.

Reviewer #2 (Remarks to the Author):

Patel et al. presented a study on the immunology of severe febrile illness. The results are well presented however they are preliminary compared to other studies published in Nature Communications which generally takes studies that generate in depth results leading to deeper insights of the disease. Therefore, the current manuscript is well suited in other journals such as Immunology or Frontiers in Immunology or similar.

Having said that the authors can improve the study to make it more fit with Nature Comm requirements by adding the following additional work to the study:

1) In immunophenotyping the authors used GLM (lines 171 - 230) to identify the themes distinct to MIS-C however their results are subjective and the validation was supported by hierarchical clustering which is not ideal. It would be better if the authors carry out deeper analysis of the RNAseq data and apply GSEA on the DEG especially the C7 geneset which has over 4000 immune related pathways then filter based on p-value and FDR and validated the top 3-5 pathways. If the top 3 pathways are activated innate immune cells then dysregulated T-cell function then mature and activated myeloid DC they can use the data they have to show that the validation is correct, but if not they should carry out additional validation on whatever the GSEA brings. This will provide deeper insights into some of the mechanisms involved in MIS in children

2) The authors need to carry out additional analysis using other software to show the role of the different immune cells. For example they can use CIBERSORT. This can be done on the targeted or the RNAseq from the blood but will provide better insights into the immune response in MIS

3) in line 638 the authors NF-kB but which part of the NF-kB they are looking for when they say activation e.g. is it p65, p50, p52 or RelA ..etc. They need to investigate this further as they may provide more interesting results because many immune cell types activate NFkB in different ways. For example recent publications showed that in pooled purified NK-T-like and NK cells (CD3+CD56+ and CD3-CD56+, respectively) the expression of mRNA coding for TLR1 to TLR9 was observed: <https://onlinelibrary.wiley.com/doi/full/10.1038/icb.2013.99>

Since TLRs are upstream of the NFkB pathway concentrating on some of the relevant NFkB molecules can provide insights into the existing findings the authors presented in this manuscript.

07 May 2024

Re: NCOMMS-23-44279-R1

Title: Immunology of severe febrile illness in children in the COVID-19 era

Reviewer #1

We thank the reviewer for taking time to critically appraise our work and providing constructive critique to improve our manuscript. We hope that our responses have addressed your concerns. We wish to note that revisions based on your comments have strengthened our work.

Major comments:

1. In their analysis of gene expression, the authors attribute a limited number of changes in mass cytometry to highly specific cellular states. However, these few proteins are not sufficiently specific for the authors' claims:

- a. For example, they use CD279 as a marker for exhausted T cells, but this gene is also upregulated on T cells in response to Type II IFNs, which does not necessarily imply exhaustion. Additionally, CD279 is upregulated on recently activated T cells and on specific subsets of T cells (eg CD4+ T follicular helper cells). Similar findings and the limitations of using CD279 as a sole marker of exhaustion have been noted in patients with MIS-C (Vella, L. et al. *Sci Immunol.*, 2021).

Response: Correlates of T cell activation, exhaustion, and apoptosis

We have run regression analyses of the co-expression of CD279 PD1 with important markers of T cell activation, exhaustion and apoptosis (HLA-DR, CD38, CD152 CTLA4, CD278 ICOS, CD134 OX40, CD95 Fas), and with intracellular IFN γ concentrations in CD4 memory T cells, CD8 memory T cells and memory regulatory T (T_{reg}) cells (Supplementary Figure 9). There was a pronounced increase in expression of CD279 PD1 in CD4 memory T cells from patients with MIS-C in comparison to other cell populations, and CD4 memory T cells from patients with other illnesses.

In general, markers of activation, exhaustion and apoptosis were highly associated ($p < 0.001$ and $R^2 > 0.20$) with CD279 PD-1 in memory CD4 T cells. In general, markers of activation, exhaustion and apoptosis had lower p values and co-efficients of determination in memory CD8 T cells. Markers of activation and exhaustion were associated with CD279 PD-1 in memory T_{reg} cells, but this was less so for CD95 Fas (used as a marker of apoptosis, $p = 0.027$ and $R^2 = 0.029$) and CD223 LAG3 (used as a marker of T cell exhaustion ($p = 0.647$, $R^2 = 0$). Thus, CD4 memory T cells co-express markers of T cell activation, exhaustion and apoptosis in severe febrile illness in children (particularly MIS-C), but this is less prominent in CD8 T cells and memory T_{reg} cells.

We now illustrate the co-expression of CD279 PD1 with markers of activation, exhaustion and apoptosis in CD4 memory and naïve/effector T cells from a representative child with MIS-C, in comparison with a representative healthy pediatric control child (Supplementary Figure 10).

Using cell stimulation / cytokine release assays, we show that there is a lower upregulation of IFN γ and other pro-inflammatory cytokines in response to SARS-CoV-2 Ag1 stimulation in children with MIS-C at T1 in comparison to vaccinated healthy adult controls, suggesting a differential T cell response to the same antigen stimulus (submitted manuscript Figure 3b). We subsequently showed that there was also a lower upregulation of IFN γ and other pro-inflammatory cytokines in response to mitogen in children with MIS-C, SBI and SVI at T1 in comparison with healthy pediatric controls and vaccinated healthy adult controls (submitted manuscript Figure 3c). These cell stimulation data support the mass cytometry data that show the co-existence of T cell activation and exhaustion in children with MIS-C and SBI.

We then analysed differential gene expression in children with MIS-C, SBI and other severe febrile illnesses (submitted manuscript Figure 5). These differential gene expression data also support the mass cytometry and cell stimulation data that show the co-existence of T cell activation and exhaustion in children with MIS-C and SBI.

We believe that our findings are in accordance with previous work describing the immunology of MIS-C¹ and with data on the immunology of adult sepsis². Vella and co-authors describe both a CD4 and CD8 T cell response in MIS-C, and use CD279 PD1 and CD39 as markers of T cell exhaustion in activated T cells. They describe a population of CX3CR1+ CD8 T cells that correlate with the use of vasoactive medication as the primary population of interest, in contrast to our data that emphasize CD4 T cells as the primary population of interest. This may represent sample timing, since Vella and co-authors sampled patients for lymphocyte cytometry typically several (≥ 5) days following presentation to hospital, in contrast to our earlier sampling (median 1.6, IQR 0.7–3.2, days for T1) and note that the CX3CR1+ CD8 T cells occur in patients with prolonged vasoactive therapy.

- b. The authors state that “Factor 2 represented restoration of innate and adaptive immune cell homeostasis, with a decrease in the proportion of immature neutrophils and increased CD11b and CD16 on myeloid cells” (lines 218 – 220). Additionally biologic proof is needed to support these statements, as these two markers are not sufficient to indicate “restoration of innate and adaptive immune cell homeostasis”

Response:

We have changed the text to include a number of other markers and changed the term for Factor 2 from “restoration of innate and adaptive immune cell homeostasis” to “restoration of innate and adaptive immune cells to baseline”. The use of MOFA reflects our wish to approach the data in an agnostic manner. However, to illustrate the additional markers used to support this statement, we have directly analysed important markers of innate and adaptive immune cell activation in patients with MIS-C and SBI at T1, T2 and T3 (Supplementary Figure 8). Data shown in these figures support the (admittedly broad) statement of restoration of innate and adaptive immune cell responses over time, features that are identified in Factor 2 in the agnostic analysis of these data.

The paragraph now reads:

Factor 2 represented restoration of innate and adaptive immune cells to baseline, with a decrease in the proportion of immature neutrophils, increased CD11b and CD16 expression and decreased phosphorylation of NF κ B in myeloid cells, and increased CD28 changes in T cells, among other changes (Supplementary Figure 8).

- c. Similarly, the author note upregulation of “genes associated with formation of NETS^{31,32}: ELANE (neutrophil elastase) and MPO (myeloperoxidase) were elevated in MIS-C in comparison with viral infection (ELANE only) and KD (Figure 5b)” (lines 341-343). Similarly, the discussion, they state: Our gene expression analysis showed increased expression of markers related to NETs in acute MIS-C, bacterial infections and KD compared with viral infections and HPC (lines 390 – 391). These genes are upregulated with neutrophil activation and are not necessarily markers of NETs. To support this statement, multiple assays (eg

bisulfite sequencing of cell-free DNA to determine cell origins, quantification of intact nucleosomes containing histone 3.1 , or citrullinated histone H3R8, among others) would need to be done.

Response

We used a series of genes that have been associated with NETosis in children with MIS-C and other severe infections in adults and children³. However, we have rephrased the paragraph noted above to reflect the uncertainty of whether upregulation of these genes represents formation of NETs, or simply upregulation of activation markers in neutrophils. We have added a small section on possible NETosis in MIS-C, SBI and KD to the *Discussion*, particularly in the light of work by Boribong and co-authors³. The relevant paragraph now reads (n.b. reference numbers correspond to this document, not the submitted manuscript):

In accordance with previous work, we observed highly activated neutrophils and other innate immune cells in MIS-C and SBI^{1,4,5}, and in KD. Neutrophil expression of FcγRs enables neutrophil phagocytosis in SBI. However, neutrophils from children with MIS-C had a distinct immune signature with extremely high expression of FcγRI and CD95/Fas. Upregulation of FcγR expression through IFNγ signaling may predispose neutrophil degranulation. Our gene expression analysis showed increased expression of genes associated with neutrophil activation, degranulation and production of NETs in acute MIS-C, bacterial infections and KD compared with viral infections and HPC. We did not, however, directly measure NET formation with microscopy and measurements of cell free DNA. However, other groups have shown extensive NET formation in MIS-C³. Formation of NETs by activated and apoptotic neutrophils may induce endothelial damage and thrombosis in MIS-C³, COVID-19 and acute respiratory distress syndrome⁶, and KD⁷. Further experiments to measure the extent of NETosis in severe febrile illness in children and to define the mechanism of NET production in severe febrile illness in children may be useful.

2. For the T cell stimulation assays, the percentages of antigen-specific T cells (eg through an AIM assay) are important for understanding the data, as this may vary greatly among the cohorts and may contribute assessments of differences in cytokine production. Intracellular staining for key cytokines (eg IFN-γ) in SARS-CoV-2-specific T cells would be more convincing evidence of the attenuated response to spike antigen that the authors report. These data are also complicated by the fact that for some of the patients with MIS-C, there was little response to mitogen stimulation (which is typically used as a positive control to indicate overall cell function despite a lack of specific Ag response).

Response: We agree that there is an attenuated response to spike antigen (SARS-CoV-2 Ag1) in the cell stimulation assays (see above) in children with MIS-C, in comparison to vaccinated healthy adult controls (note, the healthy pediatric controls were seronegative for SARS-CoV-2 IgG/IgM). However, more generally, we hypothesized that children with severe febrile illness would have a lower ability to upregulate cytokine production in response to pathogen antigen due to T cell exhaustion. The data that show little response to mitogen

stimulation in children with MIS-C, and also in children with SBI, are supportive of this hypothesis.

We have measured intracellular IFN γ in the mass cytometry experiments. We were unable to stain for T cells with the reported T cell receptor expansion (described as V β 21.3⁸ or V β 11.2^{9,10}) because the metal-conjugated antibodies to this region did not (and do not to our knowledge) exist. Such an approach would also limit the applicability of the experiment to children with MIS-C, rather than the comparative immunological approach to severe febrile illnesses (including severe bacterial infection and severe viral infection from several different pathogens) in children that we have taken.

3. The manuscript describes findings, but unfortunately cannot identify underlying mechanisms.
 - a. For example, the authors state that: “Taken together, MOFA showed differences in myeloid cell and T cell activation markers predominate differences between groups of patients with severe febrile illnesses and innate and adaptive immune cell function is restored by convalescence in MIS-C and SBI.” (lines 227 – 230). What are the mechanisms underlying these differences?

Response:

We were not able to test children *before* they become unwell, and without adequate models of disease, we are unable to accurately discriminate between pathways that are pathological or pathways that represent a beneficial host response to severe inflammation. We have extended the *Discussion* to highlight these limitations and areas for future research. We also hope that the reviewer agrees with us that testing children *before* they become unwell with acute febrile illnesses is not feasible.

The sentences of relevance now read:

A key limitation to many observational studies of childhood disease is the lack of sampling prior to illness. This limits our ability to infer differences between causal pathways to disease, and “bystander” inflammation. Further *ex vivo* experiments may be useful, particularly regarding modulation of neutrophil activation.

- b. Similarly, the differences in gene expression are described without additional context or follow-up investigations, as line lines 366 – 370 presented increased expression of IFNAR1, IFNGR1/2 in patients with MIS-C, SBI, and KD compared to those with SVI) and increased TLR3 in patients with SVI and KD compared to MIS-C and SBI. What is the significance of these findings?

Response:

We have extended the *Discussion* accordingly, noting that decreased IFN α , IFN γ 1/2 receptor expression has been previously noted in severe viral infection. It is unclear whether this represents a pathological response to viral disease or whether this is a beneficial downregulation of IFN signalling for these children to limit inflammatory damage. Similarly, we have expanded the *Discussion* section to note the significance of upregulated TLR-3 expression in viral infection. The sentences now read:

In the whole blood differential gene expression analysis we noted increased expression of receptors for IFN α/β and for IFN γ in children with MIS-C, bacterial illness and KD in comparison with viral illness, implying continued upregulation of inflammatory responses in these illnesses. It is unclear if the relative downregulation of *IFNAR1/IFNAR2/IFNGR1/IFNGR2* in SVI is a physiological response to limit over-exuberant inflammation or is a pathological process. Conversely, *TLR3* is required for clearance of herpes simplex virus¹¹, but implicated its upregulation is implicated in damaging innate and adaptive immune responses in respiratory infection¹². Future work should explore the association of IFN γ signaling with T cell exhaustion in severe febrile illnesses in children.

4. This claim needs additional clarification: “Further experiments to define the mechanism of NET production in severe febrile illness in children may be useful, particularly since polyspecific intravenous immunoglobulins are used as treatment in MIS-C, KD, and toxic shock syndrome” (lines 392 – 394). Are the authors implying that IVIG influences NETs?

Response:

There is reasonable evidence that NETs are produced in MIS-C³ and SBI², and our data here do not contradict this. We consider there to be reasonable evidence from other studies that IVIG modulates the formation of NETs that are induced by application of plasma from patients with severe COVID-19 to ex vivo healthy donor neutrophils^{6,13,14}. Given that IVIG is widely used for the treatment of MIS-C¹⁵ and some SBI (particularly streptococcal toxic shock syndrome)¹⁶, we believe that further experiments to define the mechanism of NET production may be useful. However, we have become more circumspect in the manuscript. The relevant sentence now reads:

Further experiments to measure, define mechanisms and explore immunomodulation of NETosis in severe febrile illness in may be useful.

5. Additional support is needed to support the authors’ statement that: “More generally, this characterization suggests a key signaling role for neutrophils.” Additional studies are needed to prove that neutrophils have a key signaling role in the pathogenesis of MIS-C. How do the authors differentiate between bystander activation/apoptosis of neutrophils from a driver effect of neutrophil activation?

Response: We have removed the “signalling” component of this sentence, since we are unable to differentiate between neutrophils as primary drivers of disease in comparison to a bystander effect. The sentence, “More generally, this characterization...” has now been removed entirely.

6. Please justify why for the human adult controls all were vaccinated but for human pediatric controls almost none were vaccinated. It is unclear to me why the healthy adult controls (HAC) were chosen as the rest of the cohorts are pediatric. It is not surprising that there are differences immunologically between non-ill vaccinated adults and severely ill febrile children. Some of the differences were between MIS-C and HAC

only and not with healthy pediatric controls (HPC) eg. lines 280-283. So if HPC and MIS-C are different than HAC, would it infer that it is an age-related difference given the immune systems of adults are different than children?

Response:

We agree that differences in T cell response to antigen between infants and older children have been well described. However, by the age of approximately 2–3 years children appear to produce robust antibody responses to protein and polysaccharide antigens in vaccines and pathogens^{20,21}. More specifically, as noted above, recent data have shown that SARS-CoV-2 mRNA vaccines elicit a robust antibody response in children aged 6 months to 5 years, similar to that of mRNA vaccination in adults¹⁹. The median age of children with MIS-C in the cohort was 9.6 years (IQR 5.4–13.2 years), an age when one might expect mature memory T cell responses following immunisation either by infection or vaccination¹⁸.

Furthermore, at the time of study recruitment there was a small unrepresentative cohort of vaccinated children in the United Kingdom due to a focus of vaccination campaigns on older adults. By the time of vaccine rollout for children in the UK a large proportion of children had been exposed to SARS-CoV-2 (and therefore the risk of developing MIS-C)¹⁷. We have captured this point in the discussion.

7. There are no details given in the table of patients about the HAC but the age differences in the paediatric cohorts is large. For example, KD median age is 1.4 years and there are only 8 KD patients. SVI 2.9 years median age. Whereas MIS-C, SBI and other inflammatory conditions median age is 9-10 years. Age alone may explain differences.

Response

We have now added citations to Table 1 and Supplementary Table 1 in the text of the Results about the HPC and the HAC respectively.

We agree that age may explain some of the differences between the severe febrile illnesses in the cohorts (but see above with a view on the development of T cell responses over the first 2–3 years of life). However, this does not make the comparisons less valid: severe viral infections and Kawasaki disease are typically syndromes of early childhood. Indeed, we sought to recruit children at the age where they commonly present with the range of severe febrile illnesses studied in the cohort – rather than seeking to recruit non-typical cases.

8. The authors do not note that MIS-C overlaps with KD and many MIS-C patients meet KD criteria. Sub-phenotypes within MIS-C are also important to evaluate.

Response

We agree that the clinical overlap between KD and MIS-C has been highlighted since the early case series of MIS-C (e.g.²²⁻²⁴). There is also considerable clinical overlap between MIS-C and toxic shock syndrome^{22,25}.

We have noted this in the *Introduction*: Clinically, MIS-C shares similarities with severe bacterial infection (SBI) including toxic shock syndrome (TSS), and Kawasaki disease (KD)²⁵. Both MIS-C and KD can also cause coronary artery aneurysms (CAA). MIS-C may

share immunological features with SBI and KD including a skew towards activated immature neutrophil populations, markers of neutrophil extracellular trap (NET) production, and reduced markers for antigen presentation.

We agree that sub-phenotypes within MIS-C may be important, and that these phenotypes may also be shared across diseases. Using cell population and cell expression markers we identified two distinct clusters (“sub-phenotypes”) in children with severe febrile illness (Cluster 1 and Cluster 2; Figure 2c). Children in Cluster 1 were more likely to receive vasoactive infusions than children with Cluster 2. We have also added the hierarchical clustering in children in whom no immunomodulation was given (Supplementary Figure 4) which supports the existence of sub-phenotypes in severe febrile illnesses in children, and MIS-C specifically.

9. The authors repeatedly mention "sepsis" and note a paper by Menon showing overlap of critical illness phenotypes. The relationship of the authors work to reference is not clear to me. MIS-C is a post-infectious inflammatory process that looks like sepsis. The patients with SBI likely many have sepsis and some of the SVI. The paper by Menon compared TBI, sepsis, ARDS and other critical illness phenotypes and showed overlap of these phenotypes. The authors are trying to show differences and are not emphasizing similarities.

Response:

We agree that this manuscript should be balancing a description of the similarities across severe febrile illness in children as well as the differences between them. We checked and can confirm that we do *not* have a reference with Menon as a first or senior co-author in the References. We have, however, taken the opportunity to add a reference to a recent paper by Schlapbach and colleagues³² that operationalizes the diagnosis of “sepsis” in children – that emphasizes the similarities between diagnoses that are conceptualized as “sepsis”.

We mention “sepsis” in the start of the Introduction:

Severe febrile illnesses in children requiring hospitalization and organ support, arise from diverse infections or inflammatory triggers, and overlap with the dysregulated host response that characterizes sepsis³²⁻³⁴.

And the start of the Discussion:

We compared the immunology of severe febrile illnesses in children over the trajectory of illnesses^{2,33,35}. This approach is supported by the framing of sepsis as a dysfunctional host response to infection³⁴, the successful use of immunomodulator therapies during the COVID-19 pandemic³⁶, and ongoing trials in sepsis^{2,33}. Using mass cytometry for peripheral blood immunophenotyping, cell stimulation assays and gene expression data with supervised, and unsupervised analytic approaches, we identified several shared and distinct immunological pathways in MIS-C, SBI, SVI and KD.

Minor comments

1. In the introduction, you state with certainty why the incidence of MIS-C has decreased. But it is not certain, and the authors may want to insert the word "likely" before the word "because".

Response

The sentence now reads:

Although the incidence of MIS-C has now declined, likely because of natural and vaccine-induced population immunity against SARS-CoV-2...

2. Expressing the cytokine concentration in pg/mL or ng/mL is standard convention. The baseline units in Supplemental Table 5 are not defined.

Response: The units have now been added as appropriate.

Reviewer #2

We thank the reviewer for taking time to critically appraise our work and providing constructive critique to improve our manuscript with validation using a different approach. Again, we wish to note that revisions based on your comments have strengthened our work considerably. Thank you.

Patel et al. presented a study on the immunology of severe febrile illness. The results are well presented however they are preliminary compared to other studies published in Nature Communications which generally takes studies that generate in depth results leading to deeper insights of the disease. Therefore, the current manuscript is well suited in other journals such as Immunology or Frontiers in Immunology or similar.

Having said that the authors can improve the study to make it more fit with Nature Comm requirements by adding the following additional work to the study:

1. In immunophenotyping the authors used GLM (lines 171 - 230) to identify the themes distinct to MIS-C however their results are subjective and the validation was supported by hierarchical clustering which is not ideal. It would be better if the authors carry out deeper analysis of the RNAseq data and apply GSEA on the DEG especially the C7 geneset which has over 4000 immune related pathways then filter based on p-value and FDR and validated the top 3-5 pathways. If the top 3 pathways are activated innate immune cells then dysregulated T-cell function then mature and activated myeloid DC they can use the data they have to show that the validation is correct, but if not they should carry out additional validation on whatever the GSEA brings. This will provide deeper insights into some of the mechanisms involved in MIS in children

Response

Thank you for highlighting the use of GSEA to infer immune pathways that are upregulated based on differential gene expression analysis in MIS-C (and other severe febrile illnesses in children).

We have directly measured immune cell populations, their expression of markers of activation, exhaustion and apoptosis, and their function (in cell stimulation assays). Further,

we explored the key themes in the data using multi-omic factor analysis (Supplementary Figure 5). The dimensionality reduction inherent to factor analyses are of use given we might anticipate several variables to be closely correlated. The factor analysis also combines data on the proportions of cell populations, and the expression of markers in cell populations. As discussed in the manuscript, Factor 1 represented myeloid cell activation and T cell activation, and Factor 2 represented restoration of innate and adaptive immune cells to baseline, and Factor 3 represented decreased phosphorylation of the cell signaling molecules STAT1 and STAT5 and NFκB.

As suggested we undertook GSEA, as implemented by the R packages fgsea, using the C7 geneset (<https://bioinf.wehi.edu.au/software/MSigDB/>) and interpreted by the Gene Ontology database (<https://geneontology.org/docs/go-enrichment-analysis/>). Supplementary Figure 16 now describes the GSEA for the following comparisons.

The top three pathways by % hits were:

a, MIS-C versus pHC:

1. Myeloid leukocyte activation, 82.2%
2. T cell differentiation, 81.6%
3. T cell activation, 80.1%

b, DB infection versus pHC:

1. Myeloid leukocyte activation, 88.0%
2. T cell activation, 84.4%
3. Regulation of defense response, 83.4%

c, DV infection versus pHC:

1. Innate immune response, 66.1%
2. Response to cytokine, 63.5%
3. Defence response to other organism, 63.4%

As predicted by Reviewer 2, these are highly similar to themes identified from the directly measured immune cell proportions and cell activation, exhaustion and apoptotic markers.

2. The authors need to carry out additional analysis using other software to show the role of the different immune cells. For example they can use CIBERSORT. This can be done on the targeted or the RNAseq from the blood but will provide better insights into the immune response in MIS.

Response

In this manuscript we have directly measured, using standard markers, the major immune cell populations found in peripheral blood in the Derivation (mass cytometry) cohort. We have used the whole blood transcriptomic data to explore the features of neutrophil activation and apoptosis, T cell activation, and cytokine signalling.

As suggested, we have now analyzed bulk whole blood transcriptomic data described in the manuscript in CIBERSORT (<https://cibersortx.stanford.edu>). We used the LM22 dataset

as the reference dataset, which is a signature matrix containing 22 functionally defined human immune cell subsets derived from fresh, frozen and fixed tissues and profiled by microarrays. There are no immune cell populations from children with MIS-C or other severe febrile illnesses in these data. We used 100 permutations to measure p-values associated with each population described in the reference dataset to the bulk transcriptomic data described in this manuscript.

The data are shown in Appendix 1. Following the removal of 3 outlying samples that had p values of ≥ 0.05 , we had patients in the following groups, MIS-C (n=38), definite bacterial infection (DB, n=186), definite viral infection (DV, n=137), Kawasaki disease (KD, n=136) and paediatric healthy controls (HC, n=134). The data show median correlations of MIS-C 0.73, DB 0.68, DV 0.62, DV 0.60, KD 0.71, HC 0.65.

Children with any severe febrile illness had lower CD8 T cells. Children with any severe febrile illness had resting CD4 T cells than healthy controls ($p < 0.05$ for all groups versus healthy controls), and MIS-C, DB and DV had the lower proportions than DV. In contrast, any severe febrile illness had higher memory activated CD4 T cells than healthy controls.

Children with MIS-C had lower proportions of macrophages than other severe febrile illnesses or healthy controls. Children with MIS-C and SBI had higher proportions of M0 (non-activated) and M2 (anti-inflammatory) macrophages, than children with other severe febrile illnesses and healthy controls. M1 (pro-inflammatory) macrophages were generally not identified by CIBERSORT in the blood of any children. This may be due to genuine absence, or limitations of inferring rare immune cell subsets from bulk transcriptomic data with reference to a mixed adult cell reference population.

Resting dendritic cells were also generally not identified in the blood of children with severe febrile illness or controls. Activated dendritic cells were a greater proportion of immune cell populations identified in children with SV in comparison with other severe febrile illnesses, which in turn may indicate altered cytokine signalling in children with SV. However, without single cell data that includes relatively rare populations such as dendritic cells, we cannot show this – this is a major reason why we have preferred to derive our immunological data from single cell (cytometry) methods and use gene expression analysis to explore our findings further.

As anticipated, following review of the cytometry data, inferred neutrophil proportions were higher in children with MIS-C than any other severe febrile illness or healthy controls, and higher in DB and KD than both SV and healthy controls.

3. in line 638 the authors NF-kB but which part of the NF-kB they are looking for when they say activation e.g. is it p65, p50, p52 or RelA ..etc. They need to investigate this further as they may provide more interesting results because many immune cell types activate NFkB in different ways. For example recent publications showed that in pooled purified NK-T-like and NK cells (CD3+CD56+ and CD3-CD56+, respectively) the expression of mRNA coding for TLR1 to TLR9 was observed:
<https://onlinelibrary.wiley.com/doi/full/10.1038/icb.2013.99>

Response

Since TLRs are upstream of the NFkB pathway concentrating on some of the relevant NFkB molecules can provide insights into the existing findings the authors presented in this manuscript.

We directly measured the phosphorylation status of the p65 subunit of NFkB and have now changed this in the manuscript, e.g. “and phosphorylated nuclear factor kappa B p65 (NFkB) expression in mature neutrophils” to emphasise the subunit specificity.

Appendix 1. Proportions of cells in the bulk transcriptomic data, as inferred by CIBERSORT, in whole blood from children with MIS-C, definite bacterial infection (DB), definite viral infection (DV), Kawasaki disease (KD) and pediatric healthy controls (HC). P values for pairwise-comparisons across time are using Wilcoxon rank sum testing.

Yours sincerely,

Prof Manu Shankar-Hari MB BS MSc MD FRCA FFICM PhD

REFERENCES

- 1 Carter, M. J. *et al.* Peripheral immunophenotypes in children with multisystem inflammatory syndrome associated with SARS-CoV-2 infection. *Nat Med* 26, 1701-1707, doi:10.1038/s41591-020-1054-6 (2020).
- 2 van der Poll, T., Shankar-Hari, M. & Wiersinga, W. J. The immunology of sepsis. *Immunity* 54, 2450-2464, doi:10.1016/j.immuni.2021.10.012 (2021).
- 3 Boribong, B. P. *et al.* Neutrophil profiles of pediatric COVID-19 and multisystem inflammatory syndrome in children. *Cell Rep Med* 3, 100848, doi:10.1016/j.xcrm.2022.100848 (2022).
- 4 Consiglio, C. R. *et al.* The Immunology of Multisystem Inflammatory Syndrome in Children with COVID-19. *Cell* 183, 968-981 e967, doi:10.1016/j.cell.2020.09.016 (2020).
- 5 Gruber, C. N. *et al.* Mapping Systemic Inflammation and Antibody Responses in Multisystem Inflammatory Syndrome in Children (MIS-C). *Cell* 183, 982-995 e914, doi:10.1016/j.cell.2020.09.034 (2020).
- 6 Arcanjo, A. *et al.* The emerging role of neutrophil extracellular traps in severe acute respiratory syndrome coronavirus 2 (COVID-19). *Sci Rep* 10, 19630, doi:10.1038/s41598-020-76781-0 (2020).
- 7 Yoshida, Y. *et al.* Enhanced formation of neutrophil extracellular traps in Kawasaki disease. *Pediatr Res* 87, 998-1004, doi:10.1038/s41390-019-0710-3 (2020).
- 8 Moreews, M. *et al.* Polyclonal expansion of TCR Vbeta 21.3(+) CD4(+) and CD8(+) T cells is a hallmark of Multisystem Inflammatory Syndrome in Children. *Sci Immunol* 6, doi:10.1126/sciimmunol.abh1516 (2021).
- 9 Porritt, R. A. *et al.* HLA class I-associated expansion of TRBV11-2 T cells in multisystem inflammatory syndrome in children. *J Clin Invest* 131, doi:10.1172/JCI146614 (2021).
- 10 Jackson, H. R. *et al.* Diagnosis of Multisystem Inflammatory Syndrome in Children by a Whole-Blood Transcriptional Signature. *J Pediatric Infect Dis Soc* 12, 322-331, doi:10.1093/jpids/piad035 (2023).
- 11 Zhang, S. Y. *et al.* TLR3 deficiency in patients with herpes simplex encephalitis. *Science* 317, 1522-1527, doi:10.1126/science.1139522 (2007).
- 12 Iwasaki, A. & Pillai, P. S. Innate immunity to influenza virus infection. *Nature Reviews Immunology* 14, 315-328, doi:10.1038/nri3665 (2014).
- 13 Masso-Silva, J. A. *et al.* Abrogation of neutrophil inflammatory pathways and potential reduction of neutrophil-related factors in COVID-19 by intravenous immunoglobulin. *Front Immunol* 13, 993720, doi:10.3389/fimmu.2022.993720 (2022).
- 14 Uozumi, R. *et al.* Pharmaceutical immunoglobulins reduce neutrophil extracellular trap formation and ameliorate the development of MPO-ANCA-associated vasculitis. *Mod Rheumatol* 30, 544-550, doi:10.1080/14397595.2019.1602292 (2020).
- 15 Channon-Wells, S. *et al.* Immunoglobulin, glucocorticoid, or combination therapy for multisystem inflammatory syndrome in children: a propensity-weighted cohort study. *Lancet Rheumatol* 5, e184-e199, doi:10.1016/s2665-9913(23)00029-2 (2023).
- 16 Parks, T., Wilson, C., Curtis, N., Norrby-Teglund, A. & Sriskandan, S. Polyspecific Intravenous Immunoglobulin in Clindamycin-treated Patients With Streptococcal Toxic Shock Syndrome: A Systematic Review and Meta-analysis. *Clin Infect Dis* 67, 1434-1436, doi:10.1093/cid/ciy401 (2018).

- 17 Aldridge, S. J. *et al.* Uptake of COVID-19 vaccinations amongst 3,433,483 children and young people: meta-analysis of UK prospective cohorts. *Nature Communications* 15, 2363, doi:10.1038/s41467-024-46451-0 (2024).
- 18 Carter, M. J., Blohmke, C. J. & Pollard, A. J. in *The Vaccine Book, Second Edition* (eds B. R. Bloom & P. H. Lambert) (Elsevier, 2016).
- 19 Nziza, N. *et al.* Humoral profiles of toddlers and young children following SARS-CoV-2 mRNA vaccination. *Nat Commun* 15, 905, doi:10.1038/s41467-024-45181-7 (2024).
- 20 Zhang, X., Zhivaki, D. & Lo-Man, R. Unique aspects of the perinatal immune system. *Nat Rev Immunol* 17, 495-507, doi:10.1038/nri.2017.54 (2017).
- 21 Knolle, J. *et al.* Children From the Age of Three Show a Developmental Switch in T-Cell Differentiation. *Front Immunol* 11, 1640, doi:10.3389/fimmu.2020.01640 (2020).
- 22 Whittaker, E. *et al.* Clinical Characteristics of 58 Children With a Pediatric Inflammatory Multisystem Syndrome Temporally Associated With SARS-CoV-2. *JAMA* 324, 259-269, doi:10.1001/jama.2020.10369 (2020).
- 23 Verdoni, L. *et al.* An outbreak of severe Kawasaki-like disease at the Italian epicentre of the SARS-CoV-2 epidemic: an observational cohort study. *Lancet* 395, 1771-1778, doi:10.1016/s0140-6736(20)31103-x (2020).
- 24 Feldstein, L. R. *et al.* Multisystem Inflammatory Syndrome in U.S. Children and Adolescents. *N Engl J Med* 383, 334-346, doi:10.1056/NEJMoa2021680 (2020).
- 25 Carter, M. J., Shankar-Hari, M. & Tibby, S. M. Paediatric Inflammatory Multisystem Syndrome Temporally-Associated with SARS-CoV-2 Infection: An Overview. *Intensive Care Med* 47, 90-93, doi:10.1007/s00134-020-06273-2 (2021).
- 26 McArdle, A. J. *et al.* Treatment of Multisystem Inflammatory Syndrome in Children. *N Engl J Med* 385, 11-22, doi:10.1056/NEJMoa2102968 (2021).
- 27 Son, M. B. F. *et al.* Multisystem Inflammatory Syndrome in Children — Initial Therapy and Outcomes. *New England Journal of Medicine* 385, 23-34, doi:10.1056/NEJMoa2102605 (2021).
- 28 Ouldali, N. *et al.* Association of Intravenous Immunoglobulins Plus Methylprednisolone vs Immunoglobulins Alone With Course of Fever in Multisystem Inflammatory Syndrome in Children. *JAMA* 325, 855-864, doi:10.1001/jama.2021.0694 (2021).
- 29 Ouldali, N. *et al.* Immunomodulatory Therapy for MIS-C. *Pediatrics* 152, doi:10.1542/peds.2022-061173 (2023).
- 30 Welzel, T. *et al.* Methylprednisolone versus intravenous immunoglobulins in children with paediatric inflammatory multisystem syndrome temporally associated with SARS-CoV-2 (PIMS-TS): an open-label, multicentre, randomised trial. *Lancet Child Adolesc Health* 7, 238-248, doi:10.1016/s2352-4642(23)00020-2 (2023).
- 31 Recovery Trial Group. Immunomodulatory therapy in children with paediatric inflammatory multisystem syndrome temporally associated with SARS-CoV-2 (PIMS-TS, MIS-C; RECOVERY): a randomised, controlled, open-label, platform trial. *Lancet Child Adolesc Health* 8, 190-200, doi:10.1016/s2352-4642(23)00316-4 (2024).
- 32 Schlapbach, L. J. *et al.* International Consensus Criteria for Pediatric Sepsis and Septic Shock. *JAMA* 331, 665-674, doi:10.1001/jama.2024.0179 (2024).
- 33 Maslove, D. M. *et al.* Redefining critical illness. *Nat Med* 28, 1141-1148, doi:10.1038/s41591-022-01843-x (2022).

- 34 Singer, M. *et al.* The Third International Consensus Definitions for Sepsis and Septic Shock (Sepsis-3). *JAMA* 315, 801-810, doi:10.1001/jama.2016.0287 (2016).
- 35 DeMerle, K. M. *et al.* Sepsis Subclasses: A Framework for Development and Interpretation. *Crit Care Med* 49, 748-759, doi:10.1097/ccm.0000000000004842 (2021).
- 36 Lamontagne, F. *et al.* A living WHO guideline on drugs for covid-19. *BMJ* 370, m3379, doi:10.1136/bmj.m3379 (2020).

REVIEWERS' COMMENTS

Reviewer #2 (Remarks to the Author):

Patel et al. addressed the concerns I had regarding the manuscript and the additional analysis provided further credence of the results.

One minor point is for the authors to insert the additional methods into the revised manuscript. For example the GSEA was only mentioned in the results section lines 339-340 but no mention at all in the methods section. It is advisable they add a paragraph in the "Whole blood RNA Sequencing" section starting at line 631 mentioning the source of the tool used and if they used the R version list the version. Also mention which gene sets were screened e.g. I believe they only used C7 in this case.

They should put any additional methods they applied in the methods section of the revised version. Other than that, I believe the concerns were addressed sufficiently for the manuscript to be accepted for publication.

Reviewer #3 (Remarks to the Author):

The authors have addressed the reviewers' comment with appropriate changes to the text and additional data provided. The findings and impact of the data are thoughtfully discussed within the complexities of the field and the additional details support their conclusions well. Thank you for your thorough revision.

Re: NCOMMS-23-44279-B

Title: Immunology of severe febrile illness in children in the COVID-19 era

Summary of changes to manuscript

Materials and Methods

- Addition of the methodology for the gene-set enrichment analysis contained in Supplementary Figure 16 as requested by Reviewer #2.

Reviewer #1

[No further comments.]

Response:

Thank you for your previous review.

Reviewer #2

Patel et al. addressed the concerns I had regarding the manuscript and the additional analysis provided further credence of the results.

One minor point is for the authors to insert the additional methods into the revised manuscript. For example the GSEA was only mentioned in the results section lines 339-340 but no mention at all in the methods section. It is advisable they add a paragraph in the "Whole blood RNA Sequencing" section starting at line 631 mentioning the source of the tool used and if they used the R version list the version. Also mention which gene sets were screened e.g. I believe they only used C7 in this case.

They should put any additional methods they applied in the methods section of the revised version. Other than that, I believe the concerns were addressed sufficiently for the manuscript to be accepted for publication.

Response:

We have added details of gene-set enrichment analysis (GSEA), at the suggested section of the Methods. We have also included the source of the tool (including references), the version of the tool, R version, and the signature gene set (C7) and its source. The data for the GSEA are available now online and this is shown. The R code for the analysis (and its source) is included on Dr Carter's GitHub page. The relevant section now reads:

Gene-set enrichment analysis (GSEA)³³ (Supplementary Figure 16) was undertaken on previously published data from Jackson et al.²⁹ that are available at ArrayExpress under accession E-MTAB-11671 and E-MTAB-12793. We used the R (version 4.4) implementation of gene-set enrichment analysis (*fgsea*, version 1.3)⁴², the C7 immunologic signature gene sets available from as part of the Human MSigDB Collections (<https://www.gsea-msigdb.org/gsea/msigdb/collections.jsp>), and Gene Ontology (GO) enrichment analysis. We compared gene enrichment between children with MIS-C and healthy controls, with definite bacterial infection and healthy controls, definite viral infection and healthy controls and children with MIS-C and definite bacterial infection. Code for the GSEA is available here: <https://github.com/michaeljamescarter/SIFIC>.

Reviewer #3

The authors have addressed the reviewers' comment with appropriate changes to the text and additional data provided. The findings and impact of the data are thoughtfully discussed within the complexities of the field and the additional details support their conclusions well. Thank you for your thorough revision.

Response
Thank you for your review.